EMBO
Molecular Medicine

# The critical role of the proto-oncogene c-*Kit* in TSC renal cystogenesis

Kamyar Zahedi [1,2,5], Sharon Barone [1,2,5], Marybeth Brooks [1,2], Wenzheng Zhang [3], Jane J Yu[4], Nathan A Zaidman[2] & Manoocher Soleimani [1,2 ✉]

## Abstract

The epithelium of kidney cysts in mouse Tuberous Sclerosis complex (TSC) models and TSC patients is composed of proliferating A-intercalated cells. The ablation of the *Foxi1* gene abolished renal cystogenesis in principal cell-specific *Tsc1* knockout (*Tsc1-KO*) mice. RNAseq studies comparing kidneys of *Tsc1-KO* vs. wild-type (WT) and *Tsc1/Foxi1*-double-knockout identified c-*Kit*, a tyrosine kinase receptor (RTK), as a transcript whose expression significantly increased in *Tsc1-KO* mice. Overexpression of FOXI1 in kidney M-1 cells significantly increased c-*Kit* expression levels. Kidney cystogenesis was abolished in *Tsc1-KO* mice by inactivating the c-*Kit* gene via the generation of *Tsc1/c-Kit*-double-knockout mice. The treatment of *Tsc1-KO* mice with Imatinib, a specific inhibitor of c-KIT, significantly diminished kidney cystogenesis. Renal cystogenesis was associated with ERK1/2, AKT, and RSK1-mediated phospho-inactivation of TSC2. In contrast, activation of ERK1/2, AKT, and RSK1, as well as phosphorylation of TSC2, was notably reduced in the kidneys of *Tsc1/c-Kit-dKO* mice. We propose that c-KIT is a crucial mediator of TSC renal cystogenesis and that its inhibition may constitute a novel approach for the treatment of kidney cysts in TSC.

**Keywords** Tuberous Sclerosis Complex; c-KIT; A-intercalated Cells; Cystogenesis
**Subject Categories** Molecular Biology of Disease; Urogenital System

## Introduction

TSC is a hereditary disorder caused by mutations in *TSC1* or *TSC2* genes and affects multiple organs, including the kidney, lung, and brain (Crino et al, 2006; Henske et al, 2016). The Tuberous Sclerosis Protein Complex (TSPC) is composed of TSC1, TSC2, and TBC1D7 molecules (Dibble et al, 2012). The TSC2 component is a GTPase-activating factor (GAF) whose stability is regulated by TSC1 and TBC1D7 (Dibble et al, 2012; Inoki et al, 2003). TSPC modulates the function of Ras-Homolog Enriched in Brain (RHEB) GTPase (Garami et al, 2003; Tee et al, 2003). In turn, RHEB in its GTP-bound state potentiates the activity of mTORC1 (Tee et al, 2003). A functional TSC-mTORC1 axis is vital to the regulation of cell growth and cellular response to its environment (Dibble and Manning, 2013; Fingar and Blenis, 2004). In TSC, RHEB is no longer under the inhibitory control of TSPC and, in its GTP-bound form, activates mTORC1, leading to unhindered cell growth and proliferation (Huang and Manning, 2008; Inoki et al, 2003).

In the kidney, TSC presents with benign angiomyolipomata and cysts (Lam et al, 2018; Rakowski et al, 2006). The loss or inactivation of TSC1 or 2 is the main driver of cell proliferation and tumor growth in TSC renal disease (Henske et al, 1995; Henske et al, 1996). Mice with principal cell-specific ablation of *Tsc1* or *Tsc2*, or pericyte-specific *Tsc1-KO*, develop numerous renal cysts, which are overwhelmingly comprised of A-intercalated cells (A-ICs) that express both *Tsc1* and *Tsc2* (Barone et al, 2021; Bissler et al, 2019b). A comparable expansion of A-ICs in the cystic epithelium is also seen in heterozygous *Tsc2* knockout (*Tsc2*+/−) mice (Barone et al, 2021; Bissler et al, 2019b; Onda et al, 1999). Similarly, epithelial cells lining the renal cysts of TSC patients are primarily comprised of A-ICs and express both TSC1 and TSC2 (Bonsib et al, 2016; Wilson et al, 2006). The expression of forkhead boxI1 (FOXI1), an essential transcription factor in intercalated cell development, is increased in the kidneys of mice with mutated *Tsc* genes (Barone et al, 2021). In addition, FOXI1 regulates the transcription of H+-ATPase (Kui et al, 2023), which is highly expressed in TSC cystic epithelium (Barone et al, 2021; Blomqvist et al, 2004). The ablation of the *Foxi1* gene completely abrogated the cyst formation in *Tsc1/Foxi1* double-knockout (*Tsc1/Foxi1-dKO*) mice (Barone et al, 2021). These results demonstrate the critical role of A-ICs and FOXI1 in TSC kidney cystogenesis, a process distinct from the loss of heterozygosity (LOH) that is observed in hamartomas and subependymal giant cell astrocytomas in TSC (Henske et al, 1995; Henske et al, 1997).

Receptor tyrosine kinases (RTKs), including c-KIT, a proto-oncogene, play critical roles in cell proliferation and differentiation (Robinson et al, 2000). The RTKs mediate their activity via signals transduced by ERK1/2, AKT, and p90 ribosomal S6 kinase 1

[1]Research Services, New Mexico Veterans Health Care System, Albuquerque, NM, USA. [2]Departments of Internal Medicine and Biochemistry, University of New Mexico School of Medicine, Albuquerque, NM, USA. [3]Department of Regenerative & Cancer Cell Biology, Albany Medical College, Albany, NY, USA. [4]Division of Pulmonary Care, Critical Care, and Sleep Medicine, University of Cincinnati College of Medicine, Cincinnati, OH, USA. [5]These authors contributed equally: Kamyar Zahedi, Sharon Barone. ✉E-mail: MSoleimani@salud.unm.edu

(RSK1) (Mihaylova and Shaw, 2011). The inactivation of TSC2 via its phosphorylation by ERK1/2, AKT, and RSK1 can lead to the hyperactivation of mTORC1 and unregulated cell growth (Ma et al, 2005; Manning et al, 2002; Roux et al, 2004; Wandzioch et al, 2004). These alterations may explain the anomalous behavior of cells that express both TSC1 and TSC2 in TSC lesions such as renal cysts and brain tubers (Barone et al, 2021; Bissler et al, 2019b; Bonsib et al, 2016; Henske et al, 1996).

The factors that promote cyst formation in TSC renal diseases are poorly defined. Here, we demonstrate that c-KIT is highly expressed in the cystic epithelium of various mouse models of TSC, as well as in TSC patients, and is critical in TSC renal cystogenesis.

Studies presented here identify c-KIT, an A-IC-associated RTK, as a critical signaling molecule that drives TSC renal cystogenesis. They demonstrate that the activation of c-KIT in A-ICs leads to the potentiation of protein kinases (ERK1/2, AKT, and RSK1) that phosphorylate TSC2 and inhibit TSPC function. The inactivation of TSPC and subsequent activation of mTORC1 lead to unregulated growth and proliferation

# Results

## Comparison of the renal transcriptome of *Tsc1-KO*, *Tsc1/Foxi1-dKO*, and wild-type mice

The kidneys' transcriptomes of 28-day-old *Tsc1-KO* and 45-day-old *Tsc1-KO* mice were compared to age-matched wild-type (WT) mice (Datasets EV1–2). The 28-day-old *Tsc1-KO* mice, compared to their WT counterparts, had a total of 1505 differentially expressed transcripts (DET), of which 1004 were significantly upregulated and 501 were significantly downregulated. Further analysis indicated that 893 of the upregulated and 436 of the downregulated DET coded for proteins. When contrasted with their WT counterparts, the 45-day-old *Tsc1-KO* mice had a total of 806 DET, of which 589 were significantly upregulated, and 217 were significantly downregulated. Among the latter, 578 of the upregulated and 201 of the downregulated DET coded for proteins. In addition to *Foxi1*, the expression of several known A-IC-specific transcripts, including $H^+$-ATPase subunits (e.g., *Atp6v0d2, Atp6v1c2,* and *Atp6v1g3*), *Slc4a1 (Ae1)*, ammonia transporters *(Rhcg and Rhbg)*, *Aqp6*, and *c-Kit* were significantly enhanced in the kidneys of *Tsc1-KO* mice.

To identify the molecules that may play critical roles in kidney cystogenesis in TSC, we compared the renal transcriptomes of *Tsc1-KO* mice (that present with numerous large cysts) with *Tsc1/Foxi1-dKO* mice that show complete abrogation of kidney cysts (Datasets EV1–3) (Barone et al, 2021). The expression of mRNAs specific to A-ICs that were enhanced in *Tsc1-KO* mice was profoundly downregulated in *Tsc1/Foxi1-dKO* or *Foxi1-KO* mice. It is noteworthy that the expression of *c-Kit* mRNA was significantly increased in the kidneys of *Tsc1-KO* mice and was markedly downregulated in the kidneys of *Tsc1/Foxi1-dKO* and *Foxi1-KO* mice (Fig. 1A).

Gene Ontology-Biological Process (GO-BP) enrichment analysis of day 28 and day 45 DET revealed the presence of 766 and 507 pathways with a false discovery rate of less than 0.05 (FDR < 0.05), respectively (Datasets EV4–5). These pathways included multiple associations with signal transduction and cell proliferation. The results of KEGG analysis of upregulated DET revealed the presence of 83 and 17 significantly enriched pathways (FDR < 0.05) in the kidneys of day 28 and day 45 animals, respectively. The KEGG enrichment pathways also included those that were associated with signal transduction, cell proliferation, and metabolism (Fig. EV1; Datasets EV6–7).

## Examination of the expression and localization of c-KIT in the kidneys of *Tsc1-KO* and WT mice

C-KIT is an RTK that is expressed by collecting duct A-ICs and is a known proto-oncogene with tumor-promoting properties in certain malignancies (Duffaud and Blay, 2003; Huo et al, 2005; Turner et al, 1992). The expression of *c-Kit* significantly increases in the kidneys of *Tsc1-KO* mice and is profoundly downregulated in *Tsc1/Fox1-dKO* mice. We compared the expression levels of *c-Kit* and *Foxi1* mRNA in the kidneys of WT, *Tsc1-KO*, *Tsc1/Foxi1-dKO*, and *Foxi1* mice. Northern blot analyses and the densitometric analysis of the results demonstrated a robust increase in *c-Kit* and *Foxi*1 expression in the kidneys of *Tsc1-KO* when compared to WT, *Tsc1/Foxi1-dKO*, and *Foxi1-KO* mice (Fig. 1B; upper and middle panels), confirming our RNA-seq data (Fig. 1A). Low magnification (20×) H&E images further confirm the presence of numerous large cysts in *Tsc1-KO* mice and a lack of cyst formation when compared to aged-matched *Tsc1/Foxi1-dKO* mice (Fig. 1C; middle and right panels, respectively).

In the kidneys of *Tsc1-KO* mice, c-KIT showed robust and widespread expression on the basolateral membrane of cystic epithelial cells, which also stained intensely for $H^+$-ATPase on their apical membrane (Fig. 1D; middle upper panel). In comparison, the kidneys of WT mice displayed no cysts and fewer A-ICs that stained with c-KIT and $H^+$-ATPase on the basolateral and apical aspect of their cell membrane, respectively (Fig. 1D; middle lower panel).

Next, we examined the phosphorylation and activation of c-KIT in the kidneys of *Tsc1-KO* mice compared to WT mice. Our western blot analyses (Fig. 1E) indicate that the levels of total c-KIT are much lower in WT compared to *Tsc1-KO* mice (Fig. 1E). In addition, our results indicate that the levels of phosphorylated c-KIT, which is its active form, are significantly increased in the kidneys of *Tsc1-KO* compared to WT mice (Fig. 1E). The significance of this finding is that the expression and activity of c-KIT increases during TSC renal cystogenesis. This increase is the result of augmented c-KIT levels and activity in the proliferating A-ICs in the renal cyst epithelium. To verify the activation of mTORC1, we examined the phosphorylation status of ribosomal S6 protein (S6) in the kidneys of WT and *Tsc1-KO* mice. Western blot analysis confirmed increased phosphorylated-S6 (p-S6) levels in the kidneys of *Tsc1-KO* compared to WT mice (Fig. 1F) illustrating the stimulation of mTORC1.

## c-KIT and FOXI1 expression in kidneys of $^{ECE/+}$*Tsc1-KO* and *Tsc2$^{+/-}$* mice

To exclude any possible developmental anomalies due to early ablation of *Tsc1*, we generated inducible *Tsc1* knockout ($^{ECE/+}$*Tsc1-KO*) mice where the expression of cre-recombinase and the ablation of *Tsc1* in AQP2-expressing cells of the kidney can be induced by tamoxifen (Chen et al, 2018). The tamoxifen-induced $^{ECE/+}$*Tsc1-KO* mice were examined for the development of cysts. Analysis of the time course of renal cyst development in $^{ECE/+}$*Tsc1-KO* mice shows

 

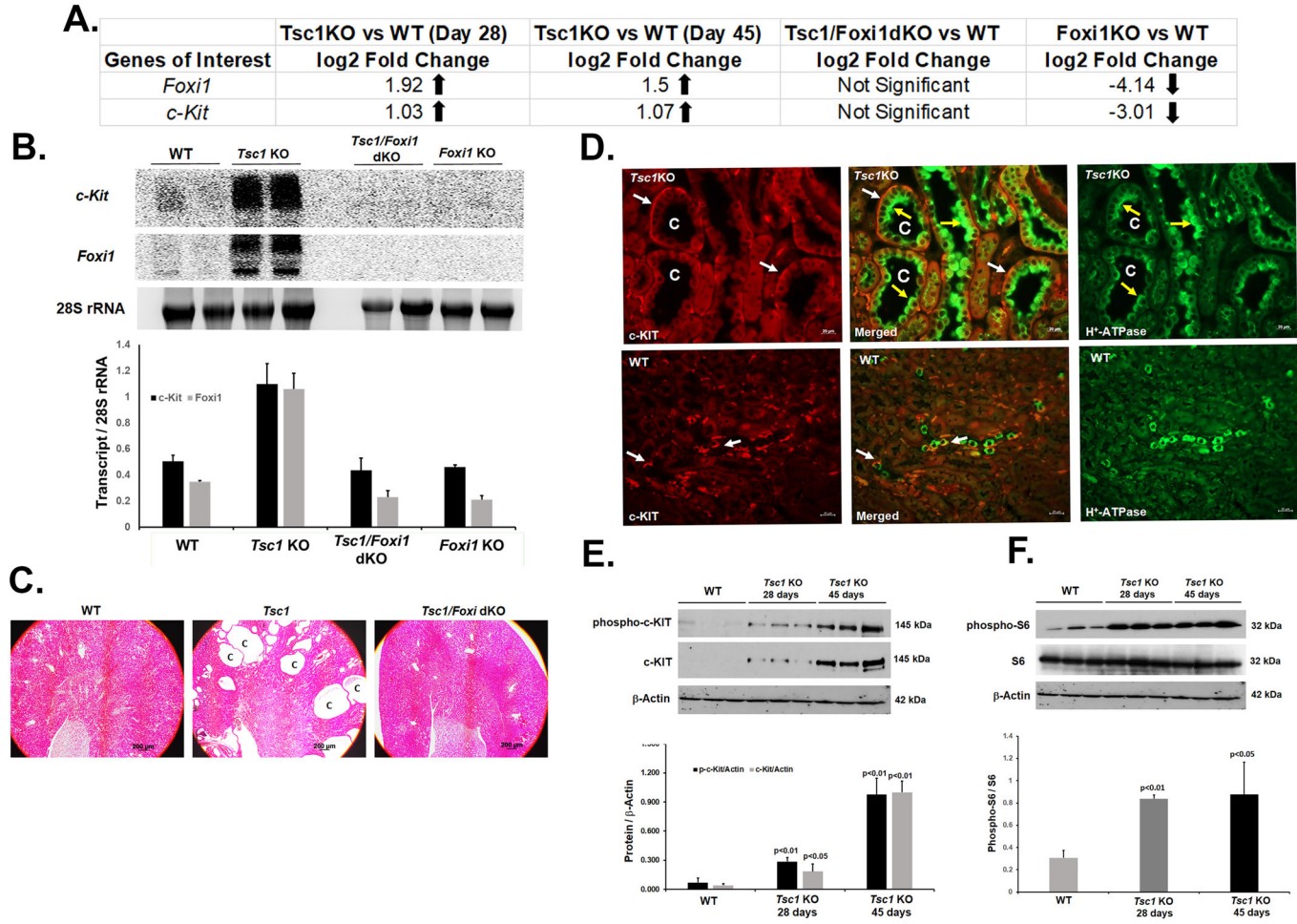

**Figure 1. Expression of c-KIT is enhanced in the epithelial lining of TSC kidney cysts.**

The expression of c-KIT was examined in the kidneys of WT, Tsc1-KO, Tsc1/Foxi1-dKO and Foxi1-KO mice and the co-localization of c-KIT and H+-ATPase in A-ICs lining the renal cyst epithelium was confirmed. (A) Table of DET of interest in the kidneys of WT, Tsc1-KO, Foxi1-KO, and Tsc1/Foxi1-dKO mice. (B) Northern blot images and Image J analysis data comparing c-Kit and Foxi1 expression in Tsc1-KO, Tsc1/Foxi-dKO and Foxi1-KO vs. WT mice. Results are presented as average ± SD. (C) Low power (4×) images of WT, Tsc1-KO, and Tsc1/Foxi1-dKO images. Scale bar equals 200 μm. "C" represents cysts. (D) Double immunofluorescence staining images of Tsc1-KO (upper panel) and WT (lower panel) mice stained with c-KIT (red staining indicated with white arrows) and H+-ATPase (green staining indicated with yellow arrows) antibodies. Scale bar equals 20 μm. "C" represents cysts. (E) Western blot analysis depicting the expression and phosphorylation of c-KIT in the kidneys of Tsc1-KO mice and WT mice ($n = 3$/group/timepoint). T-test analysis of Image J results indicate that the levels of c-KIT and phosphorylated c-KIT are significantly higher in Tsc1-KO mice at 28 days ($n = 3$; $p = 0.037237707$ and $p = 0.003699393$, respectively) and 45 days ($n = 3$; $p = 0.000159035$ and $p = 0.000880299$, respectively) compared to WT mice. Results are presented as average ± SD. (F) Western blot analysis depicting that the phosphorylation of S6 is significantly higher in the kidneys of Tsc1-KO mice. Image J analysis of the results indicates that the phosphorylation of S6 is significantly elevated in the kidneys of Tsc1-KO mice compared to WT mice ($n = 3$/group/timepoint) at 28 and 45 days of age ($p = 0.000265$ and $p = 0.030607$, respectively). Results are presented as average ± SD. Source data are available online for this figure.

that at 30 days post-tamoxifen injection demonstrates the absence of detectable renal cysts. Small and moderately sized cysts were present at 75 and 140 days after the last injection of tamoxifen, respectively, and at 230 days post-induction, mouse kidneys showed numerous large cortical cysts (Fig. 2A; 20× magnification). Northern blot analysis of kidney RNA from WT and $^{ECE/+}$Tsc1-KO mice shows a strong induction of c-Kit mRNA at 230 days (Fig. 2B), which was absent in WT and at 30 days post-tamoxifen injection in $^{ECE/+}$Tsc1-KO mice. The c-Kit mRNA levels correlated with the robust expression of Foxi1 mRNA (Fig. 2B).

Immunofluorescence labeling of the kidneys of $^{ECE/+}$Tsc1-KO mice with c-KIT and H+-ATPase antibodies showed almost universal expression of c-KIT on the basolateral membrane (Fig. 2C; left panel)

and H+-ATPase on the apical membrane of cells lining the cysts (Fig. 2C; right panel). The merged image (Fig. 2C; middle panel) clearly demonstrated the co-localization of c-KIT and H+-ATPase in the same epithelial cells lining the cysts. Double immunofluorescence labeling with H+-ATPase and AQP-2 antibodies in $^{ECE/+}$Tsc1-KO mice is shown in Fig. EV2 and demonstrates an almost universal labeling of H+-ATPase on the apical membrane of cells lining the cysts (Fig. EV2; right upper panel) with no detectable expression of AQP-2 adjacent to any H+-ATPase-expressing cells in cyst epithelia (Fig. EV2; left upper panel). The expression and localization of H+-ATPase and AQP2 in WT mice are shown for comparison (Fig. EV2; lower panels). These results along with the double immunofluorescence labeling with c-KIT and H+-ATPase antibodies (Fig. 2C) indicate that cells lining the cysts

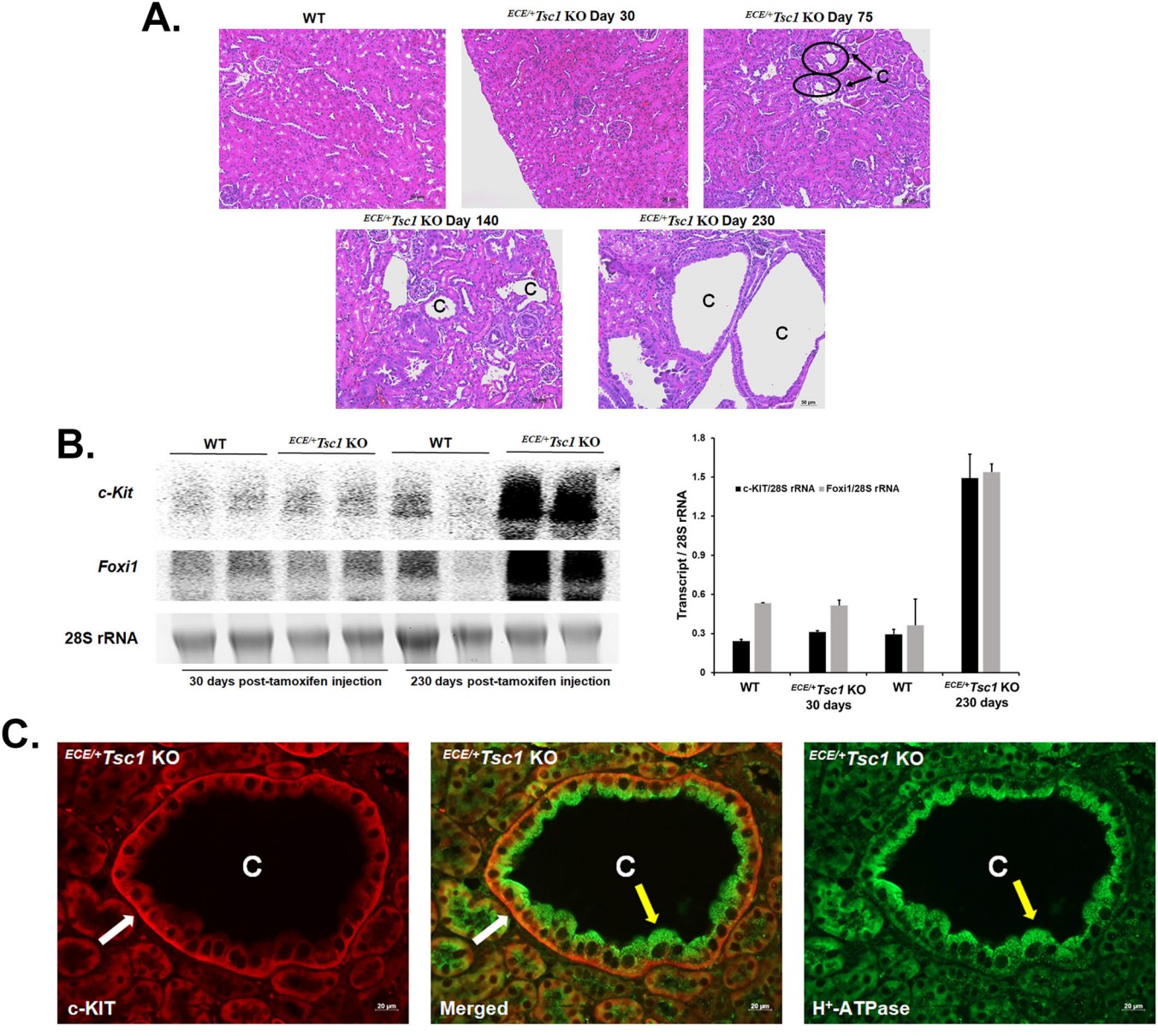

**Figure 2. Cystogenesis progression and c-KIT and FOXI1 expression in $^{ECE/+}$Tsc1 KO mice.**

(A) H&E images (20×) of tamoxifen-induced cysts in the kidneys of $^{ECE/+}$Tsc1 KO mice taken at day 30, 75, 140, and 230. Few to no cysts were present in Day 30. Small cysts start to develop by day 75 (circled) and progress to very large cysts by day 230. (B) Northern blot images and Image J analysis results comparing the expression of c-Kit (top panel) and Foxi1 (middle panel) in $^{ECE/+}$Tsc1-KO at 30 and 230 days ($n = 2$/group/timepoint) after the completion of tamoxifen treatment vs. WT mice. Results are presented as average ± SD. (C) Immunofluorescence microscopic images showing the co-localization of c-KIT (red; left panel) and H$^+$-ATPase (green; right panel) in the kidneys of WT and $^{ECE/+}$Tsc1 KO mice. The middle panel depicts a merged image of c-KIT and H$^+$-ATPase localization. White arrows point to basolateral c-KIT expression, while yellow arrows indicate apical H$^+$-ATPase localization. "C" represents a cyst. Scale bar equals 50 μm (B) and 20 μm (C). Source data are available online for this figure.

in the kidneys of $^{ECE/+}$Tsc1-KO are almost entirely composed of A-ICs that express high levels of c-KIT on their basolateral membrane.

$Tsc2^{+/-}$ mice exhibit slow development of renal lesions and cyst growth (Onda et al, 1999). Histological examination of the kidneys of $Tsc2^{+/-}$ mice shows the development of cysts by 10 months and the presence of severe lesions and large cysts by 15 months of age (Fig. 3A). Examination of the expression of c-Kit in the kidneys of $Tsc2^{+/-}$ mice, indicate a significant enhancement in c-Kit mRNA

expression in 15-month-old, but not in 6-month-old $Tsc2^{+/-}$ mice (Fig. 3B). The onset of c-Kit expression illustrates a remarkable correlation with the induction of Foxi1 in $Tsc2^{+/-}$ mice (Fig. 3B). The localization of c-KIT in the kidneys of $Tsc2^{+/-}$ demonstrates an almost uniform distribution on the basolateral membrane of cells lining the cysts (Fig. 3C; left panel). The expression of H$^+$-ATPase on the apical membrane of the same cells confirms their identity as A-ICs (Fig. 3C; right panel).

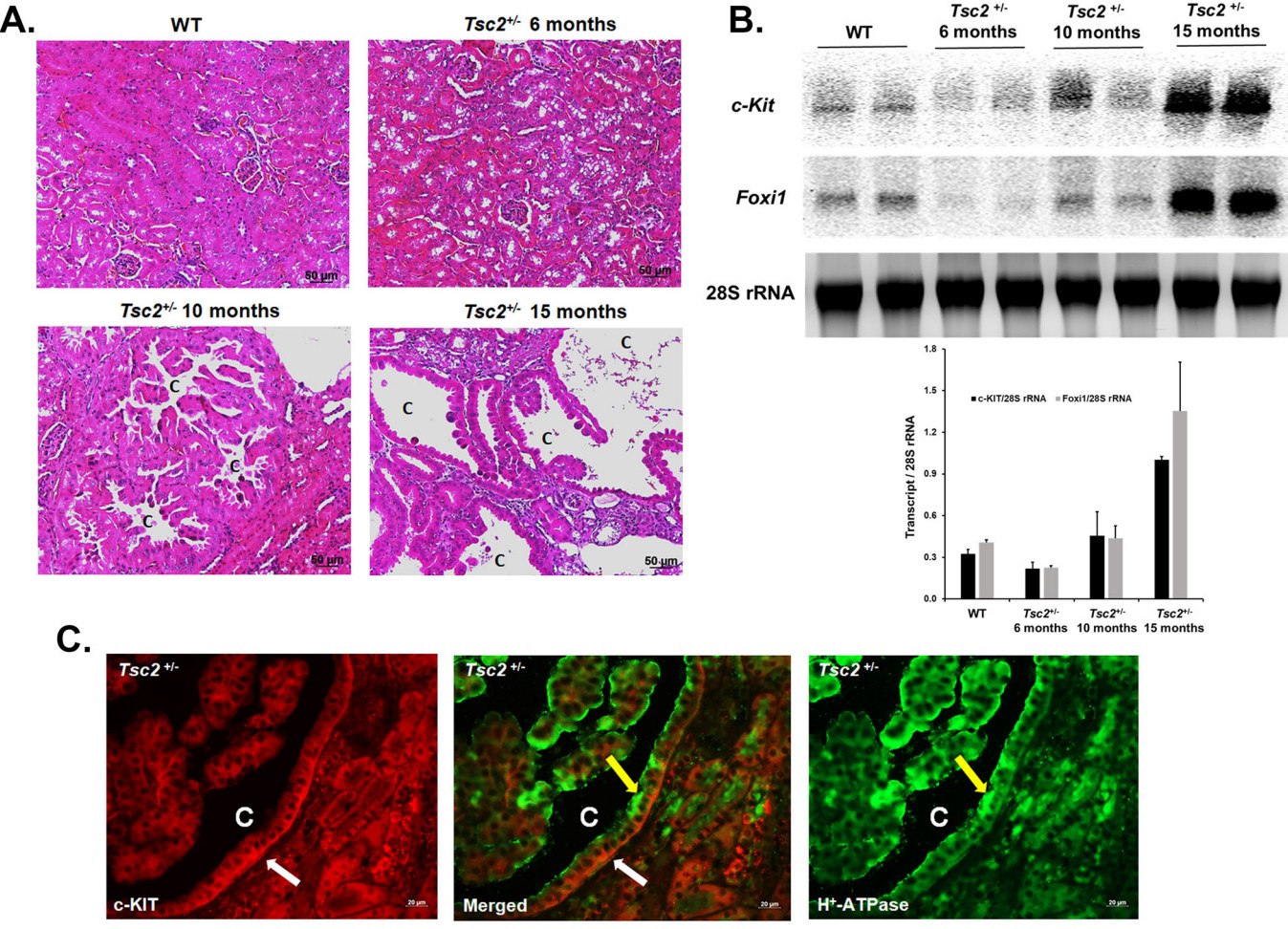

**Figure 3. Renal cystogenesis and c-KIT and FOXI1 expression in *Tsc2*$^{+/-}$ mice.**

The timeline of c-KIT expression and cystogenesis in the kidneys of *Tsc2*$^{+/-}$ mice was examined. (**A**) H&E images (20×) of the kidneys of WT and *Tsc2*$^{+/-}$ mice taken 6, 10, and 15 months of age. (**B**) Northern blot images and corresponding Image J analysis results comparing the expression of *c-Kit* (top panel) and *Foxi1* (middle panel) in the kidneys of *Tsc2*$^{+/-}$ mice at 6, 10, and 15 months of age (*n* = 2/group/timepoint). Results are presented as average ± SD. (**C**) Representative immunofluorescence microscopic images showing the co-localization of c-KIT (red; left panel) and H$^+$-ATPase (green; right panel) in the kidneys of WT and *Tsc2*$^{+/-}$ mice. The middle panel depicts a merged image of c-KIT and H$^+$-ATPase localization. White arrow points to basolateral c-KIT expression and the yellow arrow indicates the apical localization H$^+$-ATPase. "C" represents a cyst. Scale bar equals 50 μm (**B**) and 20 μm (**C**). Source data are available online for this figure.

Next, we investigated the expression of *c-Kit* mRNA in the kidneys of *Pkd1*KO mice. Unlike TSC mouse models whose cysts are comprised of A-ICs, the kidney cysts of *Pkd1*KO mice are primarily composed of principal cells (Barone et al, 2021). *Pkd1*KO mice did not demonstrate any enhancements in the expression of *c-Kit* or *Foxi1* (Fig. EV3), indicating the presence of different cystogenic mechanisms in TSC vs. ADPKD.

## Transcription factor FOXI1 enhances the expression of *c-Kit* in M-1 cells

RNA-seq results comparing the kidneys of *Tsc1-KO* vs. *Tsc1/Foxi1-dKO* mice (Datasets EV1–4) suggest that the expression of *c-Kit* is regulated by FOXI1. To determine the role of FOXI1 in the transcriptional upregulation of *c-Kit*, M-1 mouse cortical collecting duct cells were transiently transfected with the *Foxi1* expression vector (Kui et al, 2023). The comparison of control to *Foxi1*-expressing cells

revealed that the expression of *c-Kit* mRNA is significantly elevated in M-1 cells expressing *Foxi1*, but not in control M-1 cells (Fig. 4A). Next, we examined the effect of increased expression of FOXI1 and c-KIT on the growth of M-1 cells. Cells stably transfected with *Foxi1* or *c-Kit* expression vectors express elevated levels of these transcripts (Fig. EV4). Utilizing Crystal Violet cell enumeration assay to quantify the number of viable cells (Chiba et al, 1998; Feoktistova et al, 2016). Our results (Fig. 4B) indicate that the expression of FOXI1 and c-KIT led to increased proliferation of M-1 cells. The comparison of *Foxi1*, *c-Kit*, and control (empty vector) transfected cells revealed that the phosphorylation of ERK1/2 and RSK1 are enhanced, while AKT phosphorylation remains constant (Fig. 4C). Staining of kidney sections for c-KIT (Fig. 4D; left panel) and Proliferating Cell Nuclear Antigen (PCNA) (Fig. 4D; right panel) showed remarkable co-localization of these molecules in the same cells lining the cysts in the kidneys of *Tsc1-KO* mice (Fig. 4D; merged image middle panel). The results presented here, along with our previous observations in other

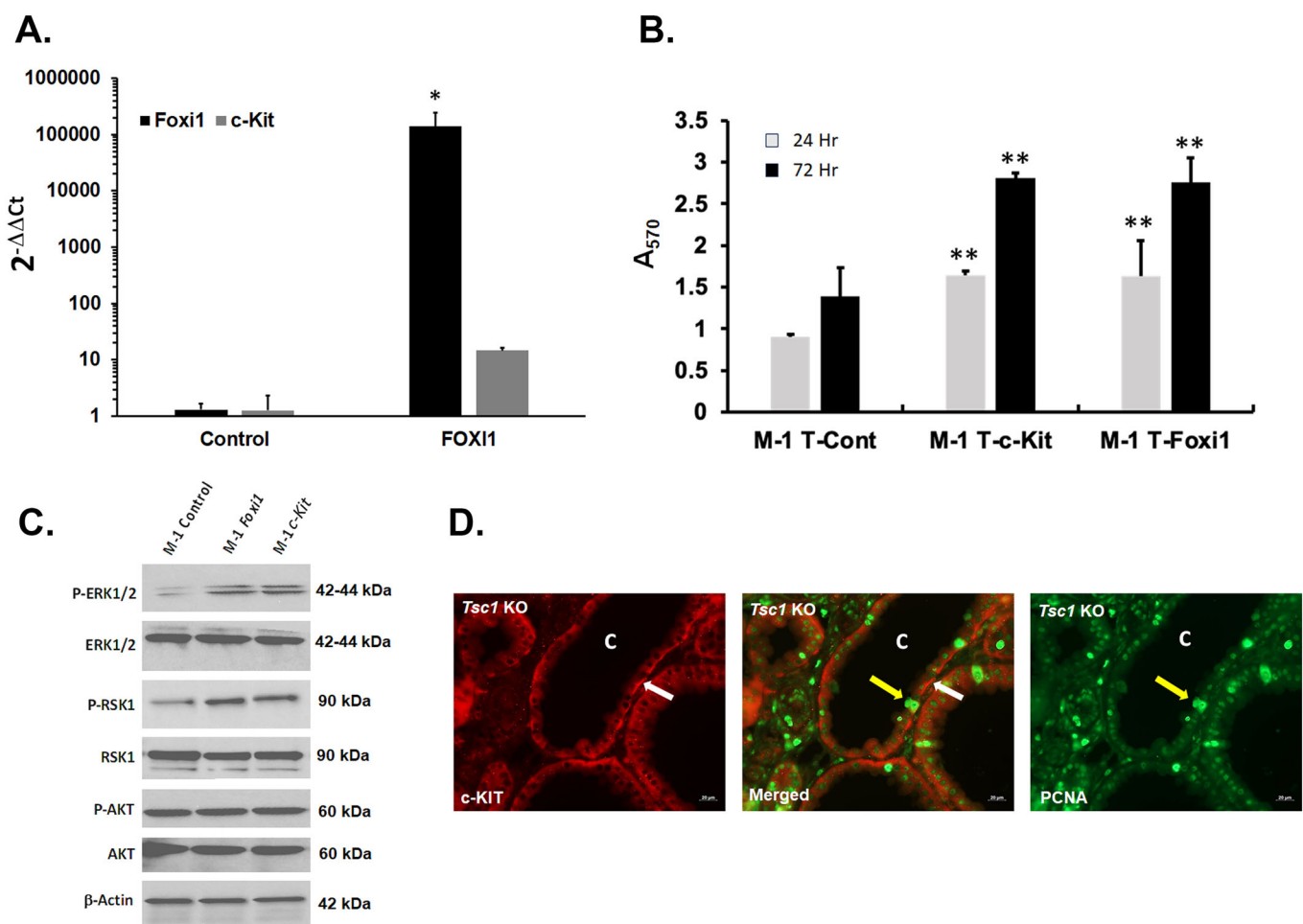

**Figure 4. Increased expression of FOXI1 and c-KIT enhances the proliferation of mouse M-1 cells.**

(A) M-1 cells transfected with *Foxi1* expression vector have significantly elevated levels of *c-Kit* transcript. The data represent the results of 3 independent transfection studies (*Foxi1* expression in Control vs *Foxi1* transfected M-1 cells using a one-tailed t-test analysis $p = 0.043411$; $*p < 0.04$. c-Kit expression in *Foxi1* vs Control transfected M-1 cells using a two-tailed t-test analysis $p = 0.000326$; $**p < 0.0003$). Results are presented as average ± SD. (B) Overexpression of Foxi1 and c-Kit (Fig. EV4) lead to increased cell proliferation (the data represent the results of 2 independent experiments ($n = 3$/cell line/timepoint/experiment) (Control vs *c-Kit* $p = 0.00290213$; Control vs Foxi1 $p = 0.00436692$; $*p < 0.01$). Results are presented as average ± SD. (C) Enhanced activation/phosphorylation of ERK1/2 and RSK1 kinases. (D) Cells lining the cystic epithelium co-express c-KIT (red staining indicated by white arrow) and PCNA (green staining indicated by yellow arrow). "C" represents cysts. Scale bar equals 20 μm. Source data are available online for this figure.

TSC models, show the co-expression of PCNA and H⁺-ATPase in cells lining the cysts (Barone et al, 2021; Bissler et al, 2019b). These results indicate that c-KIT is widely expressed in actively proliferating A-ICs that comprise the cyst epithelium.

### Expression of c-KIT in the kidneys of TSC patients

The expression of c-KIT in the cystic epithelium of biopsy samples from TSC patients was examined by immunofluorescence microscopy (Fig. EV5). Our results indicate that c-KIT expression is increased and localizes to the basolateral membrane of cells that express H⁺-ATPase on their apical membrane (Fig. EV5; left and right panels, respectively). These results confirm that the enhanced expression of c-KIT is also observed in the A-ICs that predominate the renal cystic epithelium of individuals with TSC.

### Ablation of the *c-Kit* gene blocks kidney cystogenesis in *Tsc1-KO* mice

To determine the role of c-KIT in TSC cystogenesis, *Tsc1/c-Kit* double-knockout (*Tsc1/c-Kit-dKO*) mice were generated. The development of renal cysts was compared in *Tsc1-KO* and age-matched *Tsc1/c-Kit-dKO* mice. The *Tsc1-KO* mice exhibited a substantial renal cyst burden by postnatal day 45; in comparison, renal cysts are undetectable in age-matched *Tsc1/c-Kit-dKO* mice (Fig. 5A,B). Long-term follow-up of up to 90 days confirms the continued lack of kidney cysts in *Tsc1/c-Kit-dKO* mice (Fig. 5A,B). The ablation of the *c-Kit* gene was confirmed by northern blot analysis of kidney RNA samples (Fig. 5C). Our results clearly demonstrate the absence of *c-Kit* transcript (Fig. 5C) in the kidneys of *Tsc1/c-Kit-dKO* and *c-Kit KO* mice. To determine the activation status of mTORC1 in the kidneys of *Tsc1-KO* vs *Tsc1/c-Kit-dKO*,

 

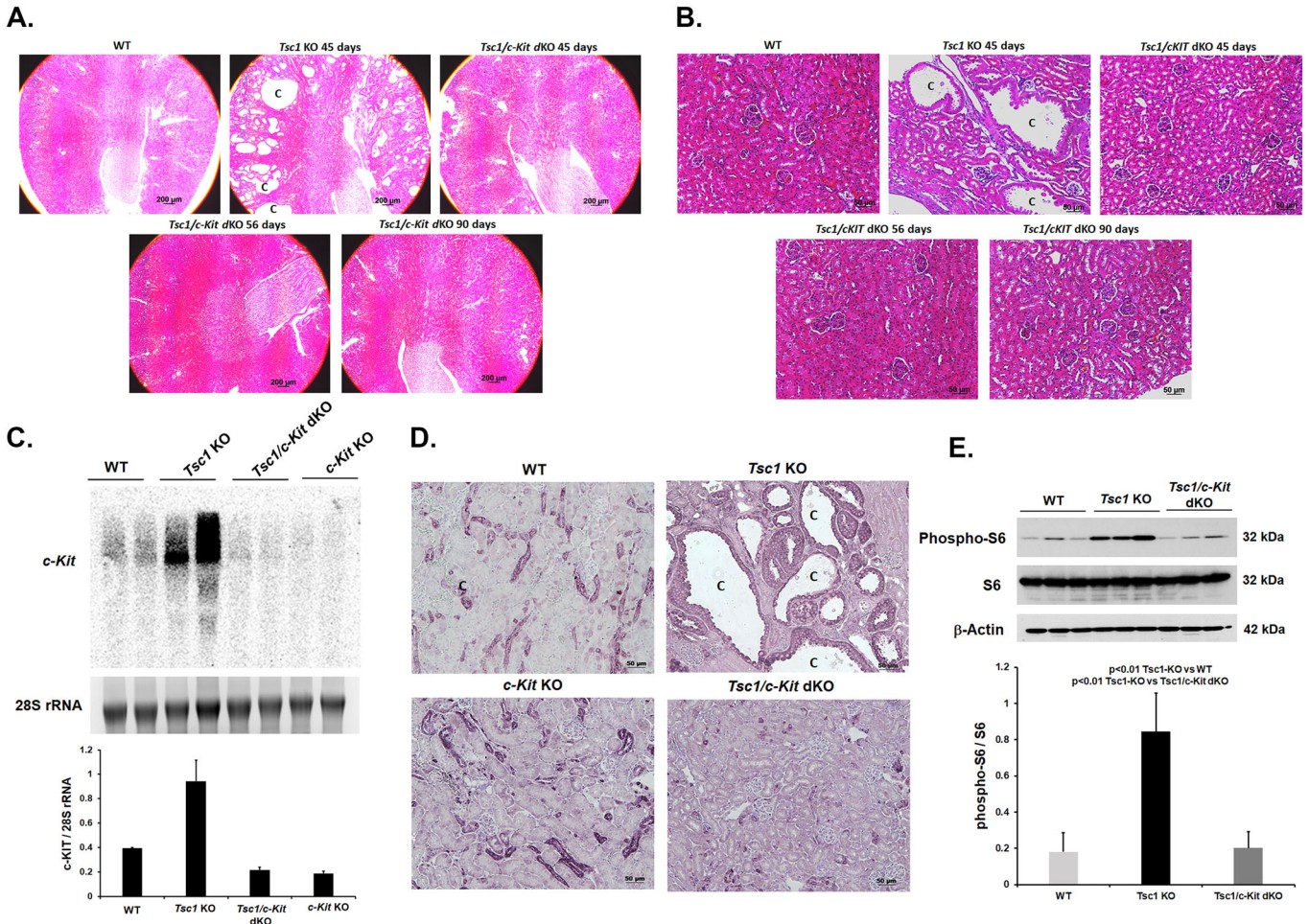

**Figure 5.  Simultaneous knockout of Tsc1 and c-Kit abolishes the renal cystogenesis.**

Comparison of renal cystogenesis in *Tsc1-KO* and *Tsc1/c-Kit*-dKO mice. (**A, B**) Representative low (4×) and high (20×) magnification H&E images of the kidneys of WT and age-matched *Tsc1-KO* and *Tsc1/c-Kit-dKO* mice indicate that the ablation of *c-Kit* gene in *Tsc1-KO* mice abrogates kidney cytogenesis. (**C**) Northern blot images and corresponding Image J analysis results examining the expression of *c-Kit* in the kidneys of WT, *Tsc1-KO*, *Tsc1/c-Kit-dKO*, and *c-Kit-KO* (mice ($n = 2$ for each strain). Results are presented as average ± SD. (**D**) Immunohistochemical staining determination of the phosphorylation of ribosomal S6 protein (p-S6) to examine the activation of mTORC1. Intense staining of p-S6 is present in cystic epithelium of *Tsc1-KO* mice (upper right panel) and significantly reduced in the kidneys of WT, *Tsc1/c-Kit-dKO*, and *c-Kit-KO* mice. Scale bar equals 200 μm (**A**), 50 μm (**B**), and 20 μm (**C**). (**E**) Western blot analysis depicting that the phosphorylation of S6 is significantly higher in the kidneys of *Tsc1-KO* compared to WT and *Tsc1/c-Kit-dKO* mice ($n = 3$/group). T-test analysis of Image J results indicates that the phosphorylation of S6 is significantly elevated in the kidneys of *Tsc1-KO* mice compared to WT ($p = 0.008413$) and *Tsc1/c-Kit-dKO* ($p = 0.008517$) mice. Results are presented as average ± SD. Source data are available online for this figure.

we examined the phosphorylation of ribosomal S6 protein (p-S6) by immunohistochemistry and western blot analysis. While *Tsc1-KO* mice showed significantly elevated levels of phospho-S6 (western blot analysis) that was localized in the cystic epithelium (immunohistochemistry), Tsc1/*c-Kit*-dKO mice had a noticeable reduction in both cyst formation (Fig. 5A,B) and mTORC1 activation (Fig. 5D,E). These results point to the critical role of c-KIT in TSC renal cystogenesis.

## Renal cystogenesis in *Tsc1-KO* mice is associated with enhanced c-KIT signaling and TSC2 inactivation

The RTK, c-KIT, is activated in the kidneys of *Tsc1-KO* mice (Fig. 1E) and enrichment analyses (Fig. EV1 and Datasets EV4–7) indicate that the MAPK and PI3K/AKT signaling pathways are

highly enriched in the kidneys of *Tsc1-KO* compared to WT mice. The RAS-MAPK and PI3K/AKT signaling pathways are utilized by RTKs, including c-KIT (Carlino et al, 2014; Regad, 2015). These signaling pathways through phosphorylation and inactivation of TSC2 are important in the regulation of the TSC/RHEB/mTORC1 axis and cell proliferation (Ma et al, 2005; Wandzioch et al, 2004). To determine if these signaling pathways are activated in the kidneys of *Tsc1-KO* mice, we compared the phosphorylation status of ERK1/2, RSK1, and AKT in the kidneys of WT mice to 28- and 45-day-old *Tsc1-KO* mice. Our results indicate that the phosphorylation of ERK1/2, RSK1, and AKT are enhanced in the kidneys of *Tsc1-KO* compared to WT mice as early as 28 days of age and remain elevated till the time of euthanasia at 45 days of age (Fig. 6A). Immunohistochemical and western blot analysis of *Tsc1-KO* kidneys indicates that the activated kinases primarily localize to

the cystic epithelium (Fig. 6B), and that p-S6 levels are elevated in in both 28 and 45 days (Fig. 1F).

Next, we examined whether TSC2, which is phosphorylated and inactivated by ERK1/2, AKT, and ERK1/2-regulated ribosomal p90 RSK1 (RSK1), is differentially phosphorylated in the kidneys of *Tsc1-KO* mice. The expression of TSC1 and TSC2 were constant and remained at similar levels in the kidneys of WT and *Tsc1-KO* mice (Fig. 6C). Comparing the phosphorylation status of TSC2 revealed that its phosphorylation on serine residues 664 (the ERK1/2 target), 939 (the AKT target), and 1798 (the RSK1 target) increase during cystogenesis (Fig. 6C). Immunofluorescence microscopic examination of $^{S1798}$p-TSC2 revealed localization to the cystic epithelium where its staining is enhanced as the animals age and cyst formation progresses (Fig. EV6).

To ascertain whether the lack of cysts in the kidneys of *Tsc1/c-Kit-dKO* mice is due to the absence of c-KIT signals that phosphorylate and inactivate TSC2, we examined the activation of ERK1/2, RSK, and AKT kinases and the phosphorylation status of TSC2 in WT, *Tsc1-KO*, and *Tsc1/c-Kit-dKO* mice. Our results indicate that the phosphorylation of all three kinases was enhanced in the cystic kidneys of *Tsc1-KO* mice. In comparison, the extent of phosphorylation of ERK1/2, RSK1, and AKT in the kidneys of *Tsc1/c-Kit-dKO* mice were significantly lower than *Tsc1-KO* and comparable with that of WT mice (Fig. 7A). Immunohistochemical analysis also revealed that the levels of phosphorylated ERK1/2, RSK1, and AKT were reduced in the kidneys of WT and *Tsc1/c-Kit-dKO* mice compared to their *Tsc1-KO* counterparts (Fig. 7B).

Comparison of TSC2 phosphorylation in the kidneys of WT, *Tsc1-KO*, and *Tsc1/c-Kit-dKO* mice (Fig. 7C) revealed that while TSC2 remained constant in WT, *Tsc1-KO*, and *Tsc1/c-Kit-dKO* mice, the levels of Ser664-, Ser939-, and Ser1798-phosphorylated TSC2 were elevated in the kidneys of *Tsc1-KO* mice compared to WT mice, whereas the phosphorylation of the same serine residues on TSC2 was reduced in the kidneys of *Tsc1/c-Kit-dKO* mice.

## The c-Kit inhibitor, Imatinib Mesylate, robustly downregulates kidney cystogenesis in *Tsc1-KO* mice

Imatinib Mesylate, a broad range inhibitor of RTKs (e.g., c-KIT and PDGFR), is used for the treatment of c-KIT-positive gastrointestinal stromal tumors, metastatic melanomas, and chromophobe renal cell carcinomas (Kim et al, 2014; Siehl and Thiel, 2007; Wei et al, 2019). We tested the effect of Imatinib Mesylate administration on renal cyst burden in *Tsc1-KO* mice. Starting at postnatal day 25, animals were treated with vehicle or Imatinib Mesylate (400 mg/kg via oral gavage) for 20 days. *Tsc1-KO* mice treated with Imatinib Mesylate displayed significantly reduced cyst burdens compared to their vehicle-treated counterparts (Fig. 8A,B). The reduction in cyst size and burden was associated with a decrease in mTORC1 activity as determined by immunohistochemical staining (Fig. 8C) and western blot analysis for p-S6 (Fig. 8D). In these studies, the Imatinib Mesylate-treated *Tsc1-KO* animals showed a reduction in the renal p-S6 levels compared to vehicle-treated *Tsc1-KO* animals (Fig. 8C,D). The kidney weights in experimental groups are shown in Fig. 8E.

## Discussion

In mouse models of TSC, as well as in individuals with TSC, the cyst epithelium continues to express both TSC1 and TSC2 (Barone et al, 2021; Bissler et al, 2019b; Bonsib et al, 2016). Similar observations have also been reported in brain tubers and some angiomyolipomata (Niida et al, 2001; Qin et al, 2010). Therefore, while LOH is the predominant mechanism behind the development of the hamartomas and subependymal giant cell astrocytomas in TSC (Crino et al, 2006; Giannikou et al, 2016), several other lesions that are associated with this disease do not conform to this mechanism (Bissler et al, 2019b; Bonsib et al, 2016; Niida et al, 2001). In TSC lesions caused by the LOH of *TSC1* gene, there is strong evidence for the destabilization of TSC2 through the compromised chaperone function of HSP70 and 90 (Woodford et al, 2017). However, in our TSC renal cyst models where the epithelial cells express both TSC1 and TSC2 (Barone et al, 2021; Bissler et al, 2019b), the LOH-dependent mechanism described by Woodford et al (Woodford et al, 2017) and Uhlmann et al (Uhlmann et al, 2002) may not be the driving force behind the formation of renal cysts. Published studies suggest that TSC kidney cystogenesis may depend on the deactivation of TSPC in cystic epithelium, leading to unregulated mTORC1 activation and cell proliferation (Barone et al, 2021; Bissler et al, 2019b; Soleimani, 2023). The lack of TSPC function in the absence of TSC1 or TSC2 deficiency may be mediated through their phosphorylation and inactivation, resulting from the potentiation of signal transduction pathways that regulate their activity. In this manuscript, we demonstrate that the enhanced expression of c-KIT by the cystic epithelial cells may drive the phosphorylation and inactivation of TSC2; thus, resulting in A-IC cellular proliferation and cyst formation.

The cystic epithelium in mouse models of TSC, as well as that of TSC cysts in humans, is almost entirely composed of A-IC (Barone et al, 2021; Bissler et al, 2019b). The expression of A-IC-associated markers, including the Cl$^-$/H$^+$ exchanger (CLC5), carbonic anhydrase 2 (CAR2), and FOXI1, is elevated in the kidneys of mouse models of TSC (Barone et al, 2023; Barone et al, 2021; Barone et al, 2024). Our studies indicate that the ablation of *Foxi1*, a gene that codes for a transcription factor that regulates the development of intercalated cells, is critical to the generation of renal cysts in TSC, and that simultaneous ablation of *Tsc1* and *Foxi1* prevents the growth of renal cysts (Barone et al, 2021). The role of the A-IC-associated molecule, CAR2, in TSC renal cystogenesis was also established in *Tsc1/Car2-dKO* mice that show delayed development of renal cysts compared to their *Tsc1-KO* counterparts (Barone et al, 2024). Comparison of renal cysts in TSC to those of ADPKD reveals that the compositions of their epithelial lining are quite different; whereas the cystic epithelium in TSC is almost entirely composed of A-IC, the ADPKD cysts are primarily made up of principal cells (Barone et al, 2021; Bissler et al, 2019b; Soleimani, 2023).

Currently, rapalogs are the only approved treatments for TSC. Although the inhibition of mTORC1 by rapalogs profoundly blunts tumor and cyst growth in TSC, the cysts and tumors return to their original size once rapalogs are discontinued (Bissler et al, 2019a). In the present study, we demonstrated that the expression and activation of c-*Kit* is significantly increased in the kidneys of *Tsc1-KO* mice (Fig. 1B,D; Datasets EV1–2). The involvement of c-KIT in renal cystogenesis and its position downstream of FOXI1 was implicated by the absence of its upregulation in kidneys of *Tsc1/Foxi1-dKO* mice that do not develop renal cysts (Fig. 1B; Dataset EV3). The absence of renal cysts in *Tsc1/c-Kit-dKO* mice

 

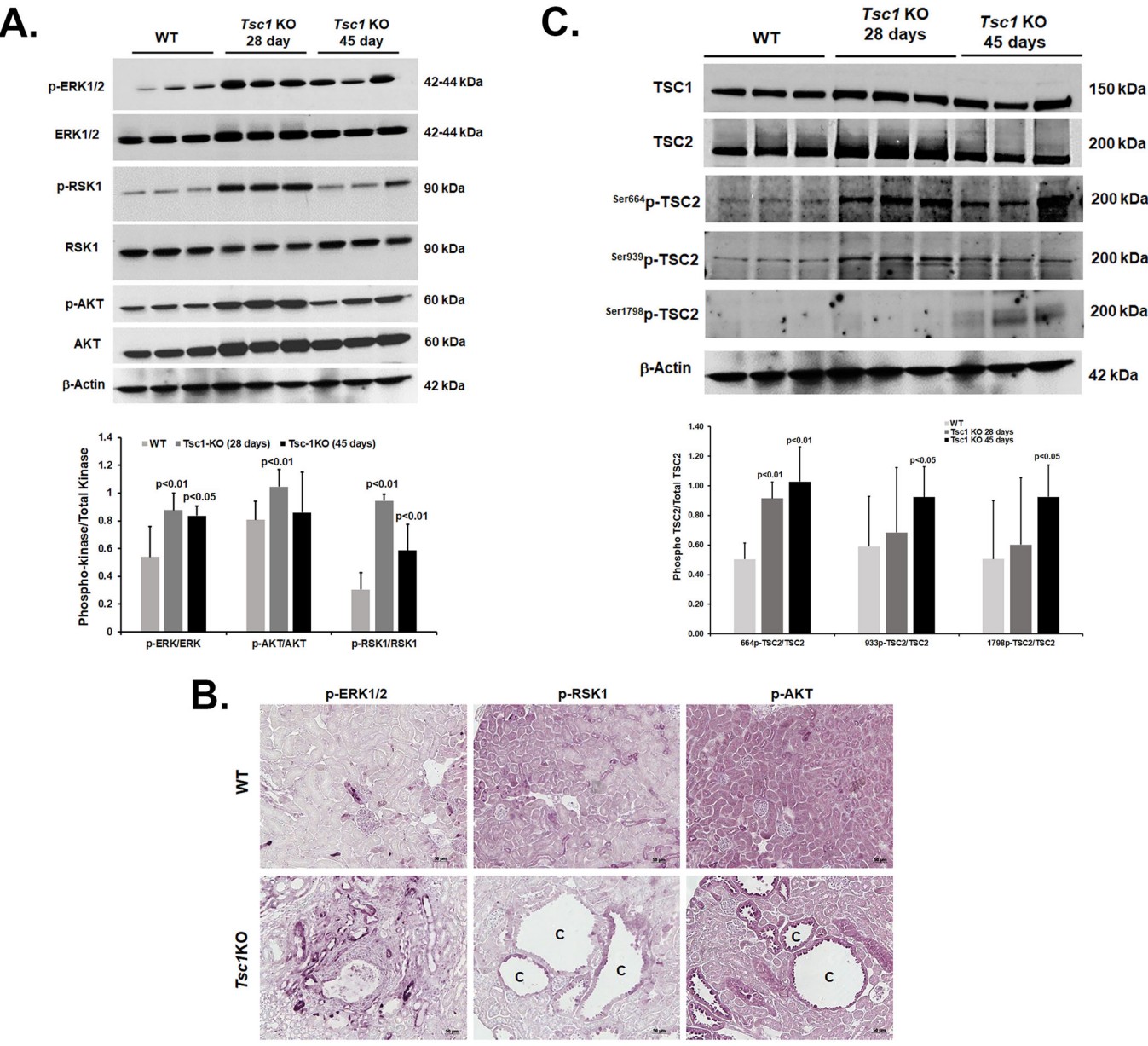

**Figure 6. RAS-MAPK signaling and TSC2 phosphorylation and inactivation in the kidneys of *Tsc1*-KO mice.**

RAS-MAPK signaling and TSC2 phosphorylation/inactivation were examined in the kidneys of WT and *Tsc1*-KO mice. (A) Phosphorylation of ERK1/2, RSK1, and AKT were compared in WT and 28- and 45-day-old *Tsc1*-KO mice ($n = 6$/group). Image J analysis of the results (histogram) indicates that the phosphorylation of all three kinases is significantly elevated in day 28 kidneys of *Tsc1*-KO compared to WT mice (pERK1/2/ERK1/2, $p = 0.009406302$; pAKT/AKT, $p = 0.007124116$; and pRSK1/RSK1, $p = 0.0003883$); while in day 45 samples only the phosphorylation of ERK1/2 and RSK1 are significantly higher in the kidneys of *Tsc1*-KO than that of WT mice (pERK1/2/ERK1/2, $p = 0.016311363$; and pRSK1/RSK1, $p = 0.00644616$). Results are presented as average ± SD. (B) Immunohistochemical labeling of p-ERK1/2, p-RSK1, and p-AKT in 45-day-old WT and *Tsc1*-KO mice. An increase in labeling can be seen within the cyst epithelium for all three phosphorylated kinases. "C" represents cysts. Scale bar equals 20 µm. (C) Phosphorylation of TSC2 by ERK1/2, AKT and RSK1 leads to its inactivation and allows unregulated mTORC1 driven cell proliferation (see the results in Fig. 1F). Phosphorylation of TSC2 on Serine residues 664, 939, and 1798 was compared in WT and *Tsc1*-KO mice. Image J analysis of the results (histogram) indicates that the phosphorylation raito of 664Serp-TSC2/TSC2 is significantly ($p = 1.78005 \times 10^{-5}$) elevated in the kidneys of *Tsc1*-KO ($n = 6$) compared to WT ($n = 8$) mice at 28 days. Image J analysis of TSC2 on day 45 indicates that all 3 sites are significantly more phosphorylated ($n = 8$; 664Serp-TSC2, $p = 5.8533 \times 10^{-5}$; 939Serp-TSC2, $p = 0.032170819$; and 1798Serp-TSC2, $p = 0.0197$) in *Tsc1*-KO ($n = 8$) compared to WT ($n = 8$) mice. Results are presented as average ± SD. Source data are available online for this figure.

and the significant reduction of the cyst burden in *Tsc1-KO* mice upon treatment with Imatinib Mesylate, confirms the critical role of c-KIT in TSC renal cystogenesis (Figs. 5A,B and 8A,B). Whether the inhibition of other RTKs, such as PDGFR, could also contribute to the effect of Imatinib Mesylate on TSC kidney cystogenesis remains to be definitively addressed (Kim et al, 2014; Siehl and Thiel, 2007). Unachukwu et al, 2023, showed that Imatinib Mesylate can reduce the severity of renal lesions in *Tsc2*$^{+/-}$ mice (Unachukwu et al, 2023). However, these studies did not directly address the role of PDGFRβ activation on the

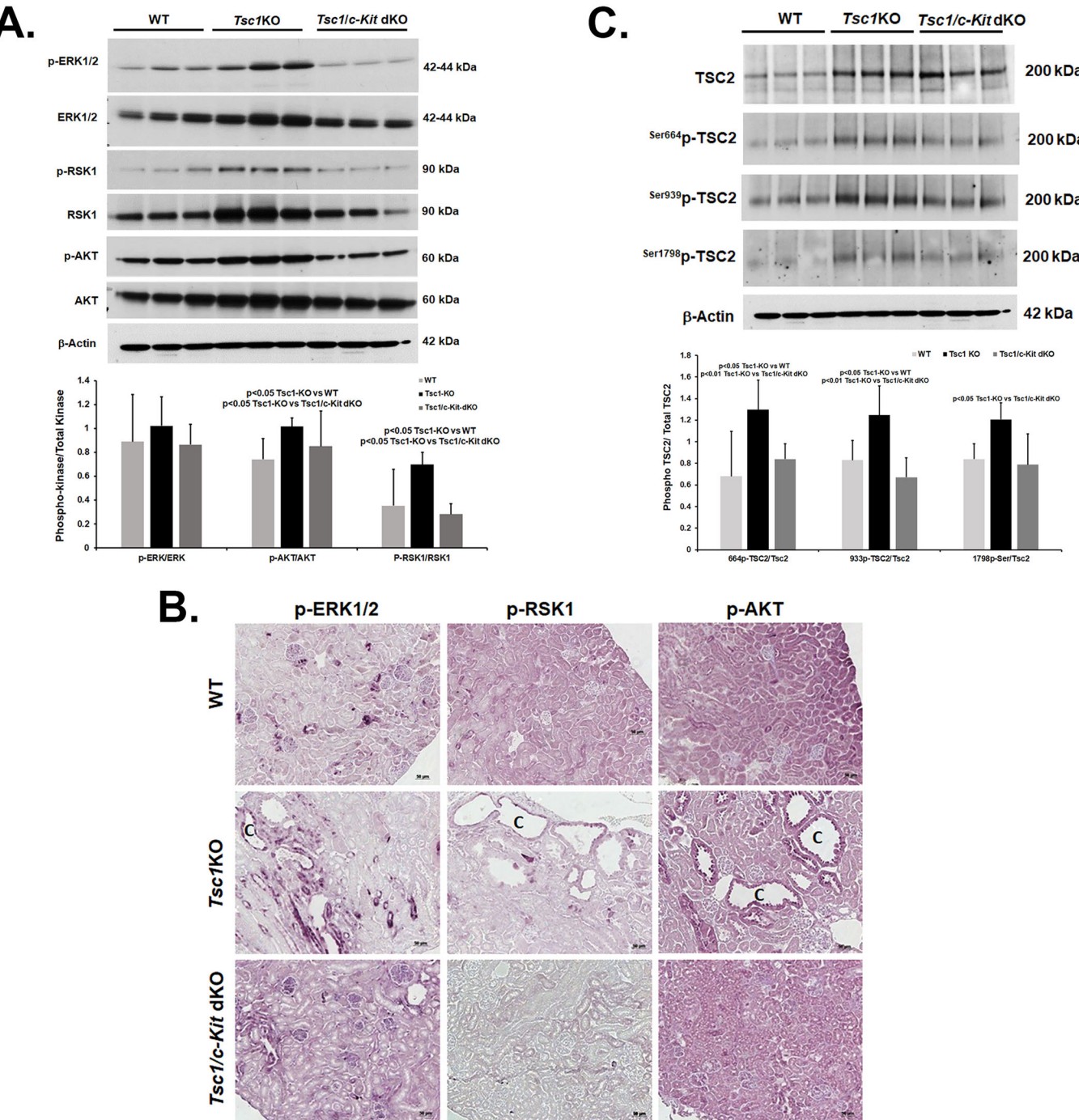

**Figure 7. Activation of ERK1/2, RSK1, and AKT signaling in time matched WT, Tsc1KO and Tsc1/c-Kit dKO mice.**

(A) Levels of phosphorylated and total ERK1/2, RSK1 and AKT were compared in WT ($n = 6$), Tsc1-KO ($n = 6$), and Tsc1/c-Kit-dKO ($n = 6$) mice. Image J analysis of phosphorylated to total activated kinases indicates that the Tsc1-KO mice have significantly elevated levels of AKT ($p = 0.010332$); and RSK1 ($p = 0.00378008$) compared to Tsc1/c-Kit-dKO mice; while comparing the phosphorylated to total activated kinase levels in Tsc1-KO and WT mice denotes significant changes in AKT ($p = 0.024352$) and RSK1 ($p = 0.006182228$) content. Results are presented as average ± SD. (B) Immunohistochemical labeling of p-ERK1/2, p-RSK1, and p-AKT in WT, Tsc1-KO, and Tsc1/c-Kit-dKO mice. "C" represents a cyst. Scale bar equals 20 μm. (C) Comparison of the renal content of phosphorylated TSC2 in WT ($n = 6$), Tsc1-KO ($n = 6$), and Tsc1/c-Kit-dKO ($n = 6$) mice was conducted by western blot followed by Image J analysis. Quantitative analysis of the results indicates that the Tsc1-KO mice have significantly elevated levels of [644Ser]p-TSC2 ($p = 0.004555628$), [939Ser]p-TSC2 ($p = 0.001481909$), and [1798Ser]p-TSC2 ($p = 0.010024844$) compared to Tsc1/c-Kit-dKO mice; while comparing the phosphorylated to total activated TSC1 levels in Tsc1-KO and WT mice indicates significant changes in [644Ser]p-TSC2 ($p = 0.012737439$) and [939Ser]p-TSC2 ($p = 0.010559358$) content. All statistics of Image J results were conducted using T-test analysis. Results are presented as average ± SD. Source data are available online for this figure.

 

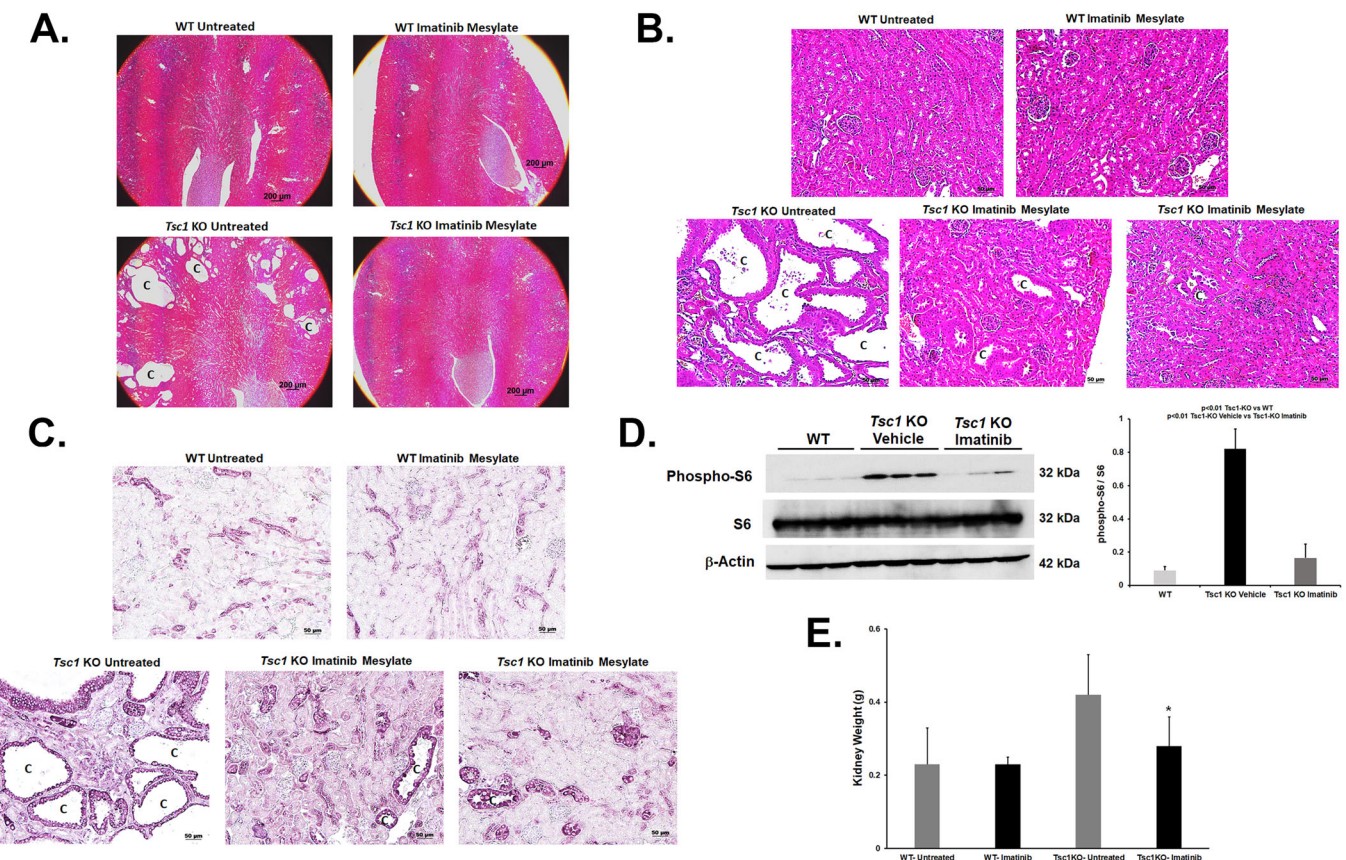

**Figure 8. Treatment with Imatinib Mesylate reduces the renal cyst burden in *Tsc1-KO* mice.**

Histology, mTORC1 activation, and kidney weights in *Tsc1-KO* mice and WT mice treated with Imatinib Mesylate were compared. (**A, B**) Low (4×) and high (20×) magnification H&E images of Imatinib Mesylate-treated and untreated *Tsc1*KO and WT mice. Treatment with Imatinib Mesylate decreased the number and size of cysts in *Tsc1-KO* mice. (**C**) Immunohistochemical staining of kidney sections for phosphorylated S6 comparing Imatinib-treated and untreated *Tsc1-KO* and WT mice demonstrate a decrease in mTORC1 activity in *Tsc1-KO* mice treated with Imatinib Mesylate. "C" represents cysts. Scale bar equals 200 μm (**A**), 50 μm (**B**), and 20 μm (**C**). (**D**) Examination of phospho-S6 levels point to significant reduction in the ratio of phospho-S6 to total-S6 in the kidneys of Imatinib-treated *Tsc1-KO* ($n = 3$) compared to vehicle-treated *Tsc1-KO* mice ($n = 3$; $p = 0.0015292$). (**E**) Comparison of kidney weight in Imatinib-treated ($n = 7$) and vehicle-treated ($n = 4$) *Tsc1-KO* and WT mice revealed that there was a significant difference (**$p = 0.045$) between *Tsc1-KO* Imatinib-treated and vehicle-treated mice highlighting the decrease in kidney cysts in the *Tsc1-KO* Imatinib-treated mice. Source data are available online for this figure.

development of TSC renal lesions (Unachukwu et al, 2023). It should be noted that while Imatinib Mesylate may be effective in the treatment of TSC lesions through the inhibition of c-KIT and PDGFRβ signaling, Imatinib is a broad inhibitor of non-receptor tyrosine kinases (e.g., BCR-Abl, c-Abl and c-Src) (Nagar, 2007; Seeliger et al, 2007; Tsutsui et al, 2016; Wang et al, 2023) and other RTK (e.g., Discoidin Domain Receptor 1; DDR1) (Bansod et al, 2021; Day et al, 2008), and may lead to unexpected off target effects. For example, inhibition of DDR1 can cause anomalies in mammary gland and arterial wound repair, as well as alterations in connective tissue structure (Curat et al, 2001; Hou et al, 2001; Vogel et al, 2001); whereas, the inhibition of c-Abl can lead to hematologic toxicity, actin reorganization, and changes in immune synapse formation (Barber et al, 2011; Huang et al, 2008).

c-KIT, through binding with the KIT ligand (KITL) regulates cell proliferation, survival, and migration (Ashman, 1999). Mutations in the c-KIT gene leading to either enhanced expression or gain of function of c-KIT can activate c-KIT signaling in affected tissues (Huo et al, 2005; Miliaras et al, 2004; Tabone-

Eglinger et al, 2008), the former results in ligand independent activation of c-KIT (Du and Lovly, 2018; Tabone-Eglinger et al, 2008), and has been implicated in the development of variety of cancers including gastrointestinal stromal tumors, melanoma, castration resistant prostate cancer, mastocytosis, and certain other hematological malignancies (Nakata et al, 1995; Tabone-Eglinger et al, 2008; Yasuda et al, 2006). c-KIT utilizes multiple signaling pathways, including the RAS-MAPK and PI3K/AKT cascades (Ma et al, 2005; Manning et al, 2002; Wandzioch et al, 2004). These pathways, when activated, participate in the regulation of TSPC function and the promotion of cell growth (Ma et al, 2005; Manning et al, 2002). As such, they may participate in the unregulated cell growth in TSC cystic lesions. Our studies demonstrated that ERK1/2, AKT, and RSK1 phosphorylation are elevated in the highly cystic kidneys of *Tsc1-KO* compared to WT and the non-cystic kidneys of *Tsc1/c-Kit-dKO* mice, indicating that RAS-MAPK and PI3K/AKT pathways are activated (Fig. 7A,B). The role of ERK1/2, AKT, and RSK1 in the inactivation of TSC2 and dysregulation of mTORC1 function is

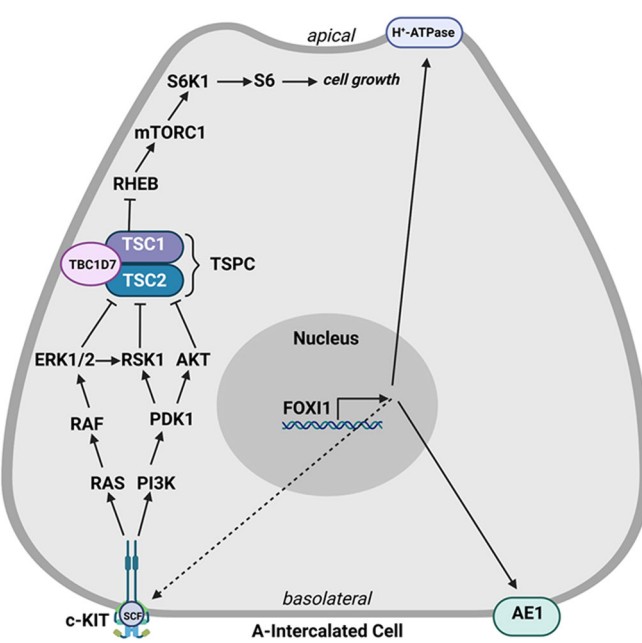

**Figure 9. Diagram of c-KIT mediated modulation of TSPC through phosphorylation and inactivation of TSC2.**

The receptor tyrosine kinase, c-KIT, is expressed on the basolateral aspect of A-IC. Activation of c-KIT leads to the potentiation of the RAS/ERK, PI3K/AKT, and RSK1 signal transduction pathways. These kinases phosphorylate and down-regulate TSC2 protein's GTPase activating function; this leads to increased stability of GTP-RHEB, unregulated mTORC1 activity, and unchecked cell proliferation. Stem cell growth factor (SCF); SLC4A1 (AE1).

well established (Ma et al, 2005; Manning et al, 2002; Roux et al, 2004; Wandzioch et al, 2004). p-ERK1/2 can directly, or through activation of RSK1, phosphorylate and inactivate TSC2 (Roux et al, 2004). AKT directly phosphorylates and inhibits TSC2. Comparison of the phosphorylation status of TSC2 in the kidneys of WT, *Tsc1-KO*, and *Tsc1/c-Kit-dKO* mice (Fig. 7C) indicates that TSC2 is phosphorylated on its ERK1/2 ($^{664}$Ser), AKT ($^{939}$Ser), and RSK1 ($^{1798}$Ser) target sites in *Tsc1-KO*, but not in *Tsc1/c-Kit-dKO* or WT mice. This signifies that the enhanced expression of c-KIT leads to phosphorylation and inactivation of TSC2 and cell expansion. Furthermore, our results indicate that genetic manipulations that reduce the severity of renal cystic disease (i.e., the ablation of *Foxi1* and *c-Kit*) can also reduce the extent of RAS-MAPK and PI3K/AKT signaling, TSC2 phosphorylation, and cystogenesis.

In addition to establishing the role of c-KIT in TSC cystogenesis, the results of our in vivo and in vitro studies described here point to a novel pathway where the increased levels of FOXI1 enhance the expression of c-KIT, leading to phosphorylation/inactivation of TSC2, unregulated activity of mTORC1, cell proliferation, and expansion. This cascade of signals may be crucial to dysregulation of cell growth in TSC cystic epithelium and explain the development of cysts in the absence of the LOH (Fig. 9). The current studies define the role of c-KIT in the development of renal cysts and establish Imatinib Mesylate as a novel treatment, either alone or in combination with rapalogs, for kidney cystogenesis in TSC.

## Methods

### Reagents and tools table

| Reagent/Resource | Reference or Source | Identifier or Catalog Number |
|---|---|---|
| **Experimental models** | | |
| *Tsc1*-KO mice | The Jackson Laboratory | 005680 |
| *Foxi1*-KO mice | The Jackson Laboratory | 024173 |
| *Aqp2* Cre mice | The Jackson Laboratory | 006881 |
| *Aqp2*$^{ECE/+}$Cre mice | Am J Physiol Renal Physiol. 2018 Apr 1;314(4):F572-F583. | Dr. Wenzheng Zhang; Albany Medical College |
| *c-Kit*-KO Mice | The Jackson Laboratory | 012861 |
| *Tsc2*$^{+/-}$ mice | J Clin Invest. 1999 Sep 15;104(6):687–695. | Dr. Jane Yu; University of Cincinnati Medical School |
| *Tsc1/Foxi1* dKO mice | Proc Natl Acad Sci USA. 2021 Feb 9;118(6): e2020190118 | Generated by Soleimani Lab |
| *Tsc1/c-Kit dKO mice* | | Generated by Soleimani Lab |
| *Pkd1* KO mice | Hum Mol Genet. 2008 Feb 7;17(11):1505–1516. | Dr. Steven Somlo; Yale University School of Medicine |
| Mouse cortical collecting duct cells (M-1 cells; Mycoplasma free) | ATCC | CRL-2038 |
| **Recombinant DNA** | | |
| Mouse c-KIT Expression Vector | Origene | MG227469 |
| Mouse Foxi1 Expression Vector | Origene | MG205758 |
| Expression Vector (Empty) | Origene | PS100010 |
| **Antibodies** | | |
| $^{Tyr719}$p-c-Kit | Cell Signaling | 3391 |
| $^{Ser664}$p-TSC2 (rabbit polyclonal) | ThermoFisher Scientific | PA5-105178 |
| $^{Ser939}$p-TSC2 (rabbit polyclonal) | Cell Signaling Technology | 3615S |
| $^{Ser1798}$p-TSC2 (rabbit polyclonal) | ThermoFisher Scientific | BS-5610R |
| ERK1/2 (rabbit polyclonal) | Cell Signaling Technology | 4695S |
| pERK1/2 (rabbit polyclonal) | Cell Signaling Technology | 4370S |
| RSK1 (rabbit polyclonal) | Cell Signaling Technology | 8408T |
| pRSK1 (rabbit polyclonal) | Cell Signaling Technology | 8753S |
| AKT (rabbit polyclonal) | Cell Signaling Technology | 4691 T |
| pAKT (rabbit polyclonal) | Cell Signaling Technology | 4060S |

| Reagent/Resource | Reference or Source | Identifier or Catalog Number |
|---|---|---|
| β-Actin (mouse monoclonal) | Santa Cruz Biotechnology | sc-47778 |
| TSC1 (rabbit polyclonal) | ProteinTech | 29906-1-AP |
| TSC2 (rabbit polyclonal) | ProteinTech | 24601-1-AP |
| p-ribosomal S6 (rabbit polyclonal) | Cell Signaling Technology | 4858S |
| S6 ribosomal (rabbit polyclonal) | Cell Signaling Technology | 2217 |
| c-KIT (rabbit polyclonal) | Cell Signaling Technology | 3074S |
| $H^+$-ATPase (mouse monoclonal) | Santa Cruz Biotechnology | sc-55544 |
| $H^+$-ATPase (rabbit polyclonal) | Generated by Soleimani Lab | |
| PCNA (mouse monoclonal) | Santa Cruz Biotechnology | sc-56 |
| AQP2 (goat polyclonal) | Santa Cruz Biotechnology | sc-9882 |
| **Oligonucleotides and other sequence-based reagents** | | |
| **Tsc1 PCR genotyping:** F4536 (5′-AGG AGG CCT CTT CTG CTA CC-3′), R4830 (5′-CAG CTC CGA CCA TGA AGT G-3′), and R6548 (5′-TGG GTC CTG ACC TAT CTC CTA-3′). | ThermoFisher Scientific | Special order oligos |
| **Aqp-2 PCR genotyping:** mAqp-2 F (5′-CCT CTG CAG GAA CTG GTG CTG G-3′) and CreTag R (5′-GCG AAC ATC TTC AGG TTC TGC GG-3′). | ThermoFisher Scientific | Special order oligos |
| **Foxi1 PCR genotyping:** Common (5′-CGA CCT CCC AGC GCC T-3′), Foxi1 WT Reverse (5′-GCT GCC TCT GCA TGC CA-3′), and Foxi1 Mutant (5′-GGC CAG CTC ATT CCT CCA CT-3′). | ThermoFisher Scientific | Special order oligos |
| **Tsc2$^{+/-}$ PCR genotyping:** H162 (5′-CAAACCCACCTCCTCAAGCTTC-3′), H163 (5′-AATGCGGCCTCAACAATCG-3′), and H164 (5′-AGACTGCCTTGGGAAAAGCG-3′). | ThermoFisher Scientific | Special order oligos |
| **Aqp2$^{ECE/+}$-Cre PCR genotyping:** Forward (5′-AAGTGCCCACAGTCTAGCCTCT-3′) and Reverse (5′-TCGCCGCTCCCGATTCGCAG-3′). | ThermoFisher Scientific | Special order oligos |
| **Foxi1 probe for Northern Hybridizations:** 5′-AGC AAG GCT GGC TGG-CAG AA-3′ and 5′-TGG CCA CGG AGC GGC TAA TA-3′ | ThermoFisher Scientific | Special order oligos |
| **c-KIT probe for Northern Hybridizations:** 5′-CATGCGTGTGTCTATGCGTG-3′ and 5′-GGGAAAACCGTGAAGGCAAC-3′ | ThermoFisher Scientific | Special order oligos |
| **TaqMan Assay Oligonucleotides:** Mm00445212_m1 (c-Kit) | ThermoFisher Scientific | Mm00445212_m1 (c-Kit) |

| Reagent/Resource | Reference or Source | Identifier or Catalog Number |
|---|---|---|
| **TaqMan Assay Oligonucleotides:** Mm00458451_m1 (Foxi1) | ThermoFisher Scientific | Mm00458451_m1 (Foxi1) |
| **Foxi1 cDNA (NM_023907.4)** | | |
| **pME18S expression vector (pFoxi1)** | | |
| **Chemicals, Enzymes and other reagents** | | |
| Tamoxifen | Sigma | T5648 |
| α-$^{32}$P-dCTP | Revity Health Services | BLU513H250UC |
| Alexa 594 goat-anti-rabbit IgG | Invitrogen | A11037 |
| Alexa 488 goat-anti-mouse IgG | Invitrogen | A11029 |
| Imatinib Mesylate | Tocris | 5906 |
| VectaStain Elite ABC Kit | Vector Labs | PK-6101 |
| Vector VIP Substrate Kit, Peroxidase | Vector Labs | SK-4600 |
| Lipofectamine 2000 | ThermoFisher Scientific | 11668019 |
| RNeasy Mini Kit | Qiagen | 74104 |
| TaqMan Gene Expression Master Mix | Applied Biosystems | 4369016 |
| **Software** | | |
| Image J | | Version 1.54p 17 February 2025 |
| Zen | Zeiss | Version 3.4 |
| DNADynamo | Blue Tractor Software | |
| GE Typhoon Scanner | GE Healthcare Life Sciences | |
| Microsoft Excel | | |
| GraphPad Prism | Dotmatics | Version 9/1/2 |
| **Other** | | |

## Animal models: generation of Tsc1-KO, ECE/+Tsc1-KO, and Tsc2+/−

All mice used in this study were treated based on the ARRIVE guidelines and the breeding and experiments set forth by our approved IACUC protocols (22-201318-B-HSC and 24-201488-HSC) at the University of New Mexico. Mice were housed at the University of New Mexico which is an AAALAC- and OLAW-accredited facility. Animals were group-housed (5 or less per cage) where water and food were provided *ad libitum*. Housing was provided at a 12-h dark/light cycle. As previously described, *Tsc1*$^{f/f}$ mice were bred with *Aqp2*-cre mice to generate *Tsc1-KO* mice, while *Tsc1/Foxi1-dKO* were generated by breeding *Tsc1-KO* mice with *Foxi1-KO* mice (Barone et al, 2021). To generate *Tsc1* Tamoxifen Inducible Cre mice ($^{ECE/+}$*Tsc1-KO*), *Tsc1*$^{f/f}$ were crossed with tamoxifen-induced *Aqp2*$^{ECE/+}$*Cre* mice (Chen et al, 2018). Tissue samples from *Pkd1* KO mice were generously provided by Dr. Stefan Somlo (Yale University School of Medicine, New Haven, CT). The TSC patient biopsy sample (Fig. EV5) was a gift provided by Dr. John J. Bissler from the University of Tennessee Health

Sciences Center (samples were obtained through an approved IRB protocol and with the full knowledge and consent of the patients).

## Genotyping protocols

Genotyping procedures for *Tsc1-KO*, *Tsc2*[+/−], *Foxi1-KO*, Aqp2Cre, and *Tsc1/Foxi1-dKO* have been described previously (Barone et al, 2021; Blomqvist et al, 2004; Onda et al, 1999). Genotyping of Cre recombinase transgene in *Aqp2*[ECE]*-Cre* was performed as previously described (Chen et al, 2018).

## Protocol for tamoxifen induction of *Aqp2 ECE/+-Cre*

Tamoxifen was dissolved in corn oil to a desired concentration of 20 μg/μl and made fresh daily. Starting at day 24, mice were given daily intraperitoneal injections of 75 mg/g body weight for 5 days. Tamoxifin and vehicle treated mice were sacrificed on days 30, 75, 140, and 230. Kidneys were harvested and snap frozen for RNA extraction or fixed and preserved for microscopy.

## Protocol for Imatinib Mesylate

Twenty-five-day-old *Tsc1-KO* and WT mice were treated with oral Imatinib Mesylate (400 mg/kg/day) or vehicle by oral gavage for 20 days. On day 45 animals were sacrificed, kidneys were harvested, weighed and snap frozen for RNA and protein extraction, or fixed and preserved for microscopy. In these studies mice that lost more than 20% of their body weight were withdrawn from the protocol and euthanized by $CO_2$ narcosis. This was a pre-established criteria based on our IACUC protocol.

## Immunofluorescence microscopy

Paraffin-embedded sections were cut in 5 μm sections and underwent antigen retrieval employing a 2100 Retriever (Electron Microscopic Sciences, Hatfield, PA). Sections were blocked in PBS containing 1% BSA, 0.2% powdered skim milk, and 0.3% Triton X-100 for at least 60 min at room temperature before incubation with primary antibodies overnight at 4 °C. Slides were washed in PBS and incubated in Alexafluor secondary antibodies (Invitrogen, Eugene, OR) at room temperature for 2 h, washed and cover-slipped. Images were obtained using a Zeiss LSM800 microscope and Zen software (version 3.4).

## RNA isolation and northern blot analysis

Mouse kidney tissues were harvested, snap frozen in liquid nitrogen, and stored in −80 °C. RNA isolation was performed using the TRI-reagent protocol. Northern blot analyses were performed as previously described (Barone et al, 2023). Membranes were hybridized at 65 °C overnight utilizing α-$^{32}$P-dCTP-radiolabeled probes for *Foxi1* (5′-AGC AAG GCT GGC TGG-CAG AA-3′ and 5′-TGG CCA CGG AGC GGC TAA TA-3′) or *c-Kit* (5′-CATGCGTGTGTCTATGCGTG-3′ and 5′-GGGAAAACCGTGAAGGCAAC-3′). Membranes were washed and Northern blot images were developed using a GE Typhoon Scanner (GE Healthcare Life Sciences).

## RNA-seq analysis

The RNA-seq analyses were performed by Novogene Bioinformatics Technology Co., Ltd (Beijing, China). Briefly, total kidney RNA was isolated as described in the previous section. The isolated RNA samples were ethanol precipitated, dissolved in water and subjected to quality control analysis using an Agilent 2100 Bioanalyzer with RNA 6000 Nano Kits (Agilent, USA). The redissolved samples were subjected to poly A selection, fragmented and reverse-transcribed to generate complementary DNA libraries that were utilized for sequencing analysis. Libraries were sequenced using the HiSeqTM 2500 system (Illumina). Clean reads were aligned to a mouse refence genome using Hisat2 v2.0.4. Gene. The gene expression levels were determined using fragments per kilobase of transcript per million mapped fragments (FPKM) by HTSeq v0.9.1. The enrichment analysis of DET were performed using ShinyGO application (http://bioinformatics.sdstate.edu/go/, 02-19-2024).

## Protein extraction and western blot analysis

Protein extracts were prepared from isolated kidneys, size-fractionated by SDS/PAGE and transferred to nitrocellulose filters. Western blot analyses were performed using antibodies that recognize proteins of interest (for more information please refer to the "Antibodies" section). Western blot images that share the same β-Actin loading control—Fig. 1E, F, Figs. 6A, C as well as Figs. 5E, 7A and 7C—originate from the same Western blot membranes.

## Image J analysis of northern and western blots

The quantification of northern blot analysis results was performed by measuring the intensity of the bands of interest (specific visualized mRNA band) in each lane and normalizing the results to that of the intensity of the corresponding 28s rRNA band. The quantification of western blot analysis results was accomplished by determining the intensity of the visualized band of interest and normalizing the result to that of the density of the corresponding β-actin band.

## Antibodies

The following antibodies were used for Western blot experiments: (1) $^{tyr719}$p-c-KIT (1:1000) (Cell Signaling Technology, Danvers, MA); (2) $^{ser664}$p-TSC2 (1:1000) (Invitrogen, Rockford, IL); (3) $^{ser939}$p-TSC2 (1:1000) (Cell Signaling Technology, Danvers, MA); (4) β-actin (1:1000) (Santa Cruz Biotechnology, Dallas, TX); (5) TSC1 (1:2000) (Protein Tech, Rosemont, IL); (6) TSC2 (1:2000) (Protein Tech, Rosemont, IL); (7) ERK1/2--WB (1:1000) (Cell Signaling Technology, Danvers, MA); (8) RSK1 WB (1:1000) (Cell Signaling Technology, Danvers, MA); (9) AKT WB (1:1000) (Cell Signaling Technology, Danvers, MA); and (10) S6 ribosomal (1:1000) (Cell Signaling Technology, Danvers, MA).

The following antibodies were utilized for immunofluorescence (IF) or immunohistochemistry (IHC) and Western blots (WB): (1) p-ribosomal S6--IHC (1:100) and WB (1:1000) (Cell Signaling Technology, Danvers, MA); (2) c-KIT antibody--IF (1:100) and WB

 

**The paper explained**

**Problem**

Tuberous Sclerosis Complex (TSC) is an autosomal dominant disease caused by mutations in hamartin (TSC1) or tuberin (TSC2) genes. It affects multiple organs, and in the kidney, presents as hamartomas and cysts. Renal cysts in TSC develop in the kidney cortex and are primarily composed of A-intercalated cells that express both TSC proteins. They are a significant source of morbidity and mortality by causing end-stage kidney failure and hypertension.

**Results**

Here, we demonstrate that the receptor tyrosine kinase, c-KIT, is overexpressed and plays a critical role in mouse TSC renal cystogenesis. Our studies specifically illustrate that the ablation or inhibition of c-KIT robustly reduces cyst burden, identifying a new approach for treating TSC renal lesions. Our mechanistic studies demonstrate that the phosphorylation and inactivation of TSC2 by c-KIT promote renal cyst formation without loss of heterozygosity (LOH), which may explain the basis of TSC pathologies in the kidney.

**Impact**

Our studies strongly suggest that c-KIT plays a crucial role in TSC renal cystogenesis, and its inhibition may offer a novel treatment for TSC renal cysts.

(1:1000) (Cell Signaling Technology; Danvers, MA); (3) pERK1/2 IHC (1:300) and WB (1:1000) (Cell Signaling Technology, Danvers, MA); (4) pRSK1 IHC (1:100) and WB (1:1000) (Cell Signaling Technology, Danvers, MA); (5) $^{ser1798}$p-TSC2 IF (1:75) and WB (1:800) (ThermoFisher Scientific, Eugene, OR); and (6) pAKT IHC (1:50) and WB (1:1000) (Cell Signaling Technology, Danvers, MA).

The following antibodies were used for immunofluorescence studies: (1) Monoclonal H$^+$-ATPase (1:50; Santa Cruz Biotechnology); (2) H$^+$-ATPase (1:75; our lab); (3) PCNA (1:50; Santa Cruz Biotechnology); and (4) AQP2 (1:50; Santa Cruz Biotechnology).

## M-1 transfection and RT-PCR analysis

M-1 cells derived from the renal cortex of a SV40 early region transgenic mouse were purchased from American Type Culture Collection (ATCC, Manassas, VA). Cells were grown in a 1:1 mixture of Dulbecco's modified Eagle's medium and Ham's F12 medium with 2.5 mM L-glutamine adjusted to contain 15 mM HEPES, 0.5 mM sodium pyruvate and 1.2 g/L sodium bicarbonate supplemented with 0.005 mM dexamethasone, 10% fetal bovine serum, 1% P/S and 5 µg/mL plasmocin (Invivogen, San Diego, CA). M-1 cells were grown on Transwell 0.4 µm pore polyester membranes (Corning, AZ). Once cultures reached 60% confluence, cells were reverse transfected with Lipofectamine 2000 (Thermo Fisher Scientific, Waltham, MA) at a plasmid mass to transfection reagent volume ratio of 2:1 and grown for at least two days prior to experimentation. *Foxi1* (NM_023907.4) cDNA was cloned from whole mouse kidney into a pME18S expression vector (pFoxi1). RNA was isolated from cells using the RNeasy Mini Kit (Qiagen, Hilden, Germany). cDNA was generated using the QuantiTect reverse transcription Kit with genomic DNA

Wipeout (Qiagen, Hilden, Germany). TaqMan real-time PCR assays were performed in triplicate according to the manufacturer's standard protocol (Thermo Fisher Scientific, Waltham, MA). Briefly, 10 ng cDNA was PCR-amplified using an Applied Biosystems QuantStudio 5 PCR system (Thermo Fisher Scientific, Waltham, MA). Amplification data was analyzed using Microsoft Excel and plotted with GraphPad Prism version 9.1.2. All qPCR data are mean values of three biological replicates and is presented as relative expression per *Gapdh*. TaqMan assays oligonucleotides are as follows: Mm00445212_m1 (*c-Kit*) and Mm00458451_m1 (*Foxi1*).

## Cell enumeration assay

M-1 cells (ATCC, Manassas, VA, USA) adapted to grow in serum-free medium (M-1 SFM) were used in these studies. The viability and proliferation of M-1 SFM cells stably transfected with Control (empty), *Foxi1* and *c-Kit* expression vectors was determined using Crystal Violet uptake assay (Chiba et al, 1998; Feoktistova et al, 2016). Briefly, $1.0 \times 10^5$ cells were seeded per well in 12-well plates. Cells were allowed to attach for 24 h. After the cell attachment period, at 24 and 72 h the cells were washed with PBS (5X), stained with 0.25% Crystal Violet solution (0.125 g Crystal Violet dissolved in 20% methanol) for 30 min at room temperature, washed with PBS (5X) and lysed with 0.1% SDS for 60 min at room temperature. The absorbance of the lysate (uptake and retention of crystal Violet) was measured at 570 nm.

## Statistical analysis

The results of experiments are presented as mean ± standard deviation (SD). The specific details of the biological and technical replicas are provided in each figure legend. The significance of results was determined using Student T test. A "*p*" value of <0.05 was considered statistically significant. This study did not include any blinding of samples.

## Data availability

The RNA-SEQ data (Datasets EV1–3) have been submitted to the NCBI-Gene expression omnibus (GEO) repository. RNASeq for Tsc1 KO vs WT Day 28; GEO database: GSE311308 https://www.ncbi.nlm.nih.gov/geo/query/acc.cgi?acc=GSE311308. RNASeq for Tsc1 KO vs WT Day 45; GEO database: GSE311309 https://www.ncbi.nlm.nih.gov/geo/query/acc.cgi?acc=GSE311309. RNASeq for Tsc1/Foxi1 dKO vs WT; GEO database: GSE311307 https://www.ncbi.nlm.nih.gov/geo/query/acc.cgi?acc=GSE311307.

The source data of this paper are collected in the following database record: biostudies:S-SCDT-10_1038-S44321-025-00360-x.

## Peer review information

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

## Acknowledgements

These studies were supported by Merit Review Award 5I01BX001000-06 (Department of Veterans Administration) (MS). Dialysis Clinics Inc. Grant C-4149 (MS). Department of Defense grants: TS240031 (MS), W81XWH2010736 and W81XWH1910474 (JJY). NIH Grants: NHLBIT32HL007736 (T. Resta, PI; MS), R01HL138481 (JJY), 1R01DK136554-01 (WZ), and 4R00DK127215-03 (NZ). MS is a Senior Clinician Scientist Investigator with the Department of Veterans Health Administration. This research made use of the Fluorescence Microscopy and Cell Imaging Shared Resource, which is supported partially by the University of New Mexico (UNM) Comprehensive Cancer Center Support Grant NCIP30CA118100. The funders had no role in study design, data collection, analysis, decision to publish, or preparation of this manuscript.

## Author contributions

**Kamyar Zahedi**: Conceptualization; Data curation; Formal analysis; Validation; Investigation; Visualization; Methodology; Writing—review and editing. **Sharon Barone**: Data curation; Formal analysis; Investigation; Visualization; Methodology; Writing—review and editing. **Marybeth Brooks**: Maintenance of transgenic mouse colonies and genotyping. **Wenzheng Zhang**: Resources. **Jane J Yu**: Resources. **Nathan A Zaidman**: Formal analysis; Investigation. **Manoocher Soleimani**: Conceptualization; Resources; Formal analysis; Supervision; Funding acquisition; Validation; Investigation; Visualization; Writing—original draft; Project administration; Writing—review and editing.

Source data underlying figure panels in this paper may have individual authorship assigned. Where available, figure panel/source data authorship is listed in the following database record: biostudies:S-SCDT-10_1038-S44321-025-00360-x.

## Disclosure and competing interests statement

The authors declare no competing interests.

# Expanded View Figures

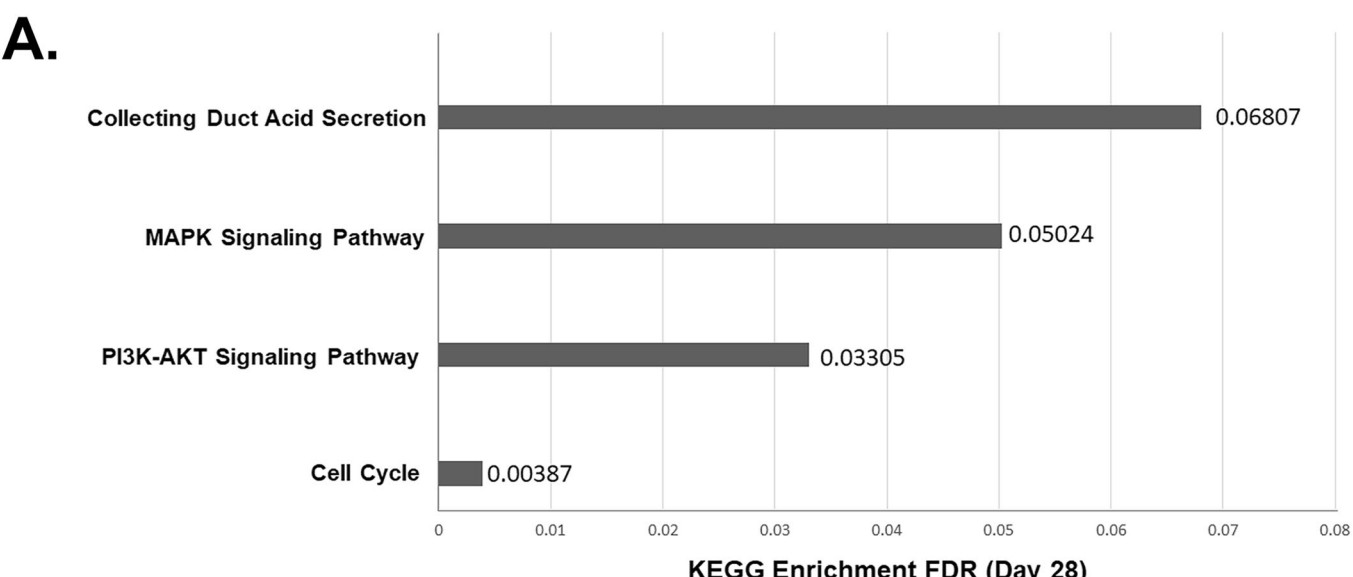

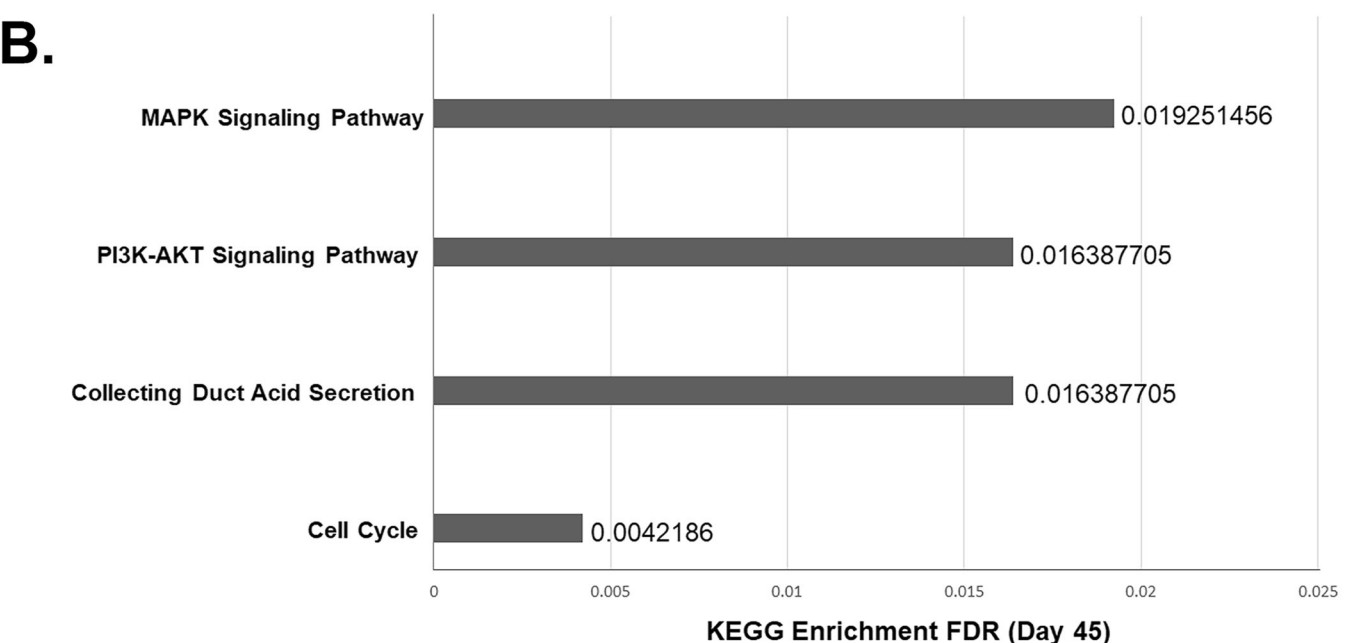

**Figure EV1. KEGG enrichment analysis.**

KEGG enrichment analysis was performed on (**A**) 28-day *Tsc1-KO* mice and (**B**) 45-day *Tsc1-KO* mice using DEG with fold induction of greater than 1.3 and FDR < 0.05. The results indicate that collecting duct acid secretion, MAPK, PI3K-AKT, and cell cycle pathways were significantly enriched. Source data are available online for this figure.

 

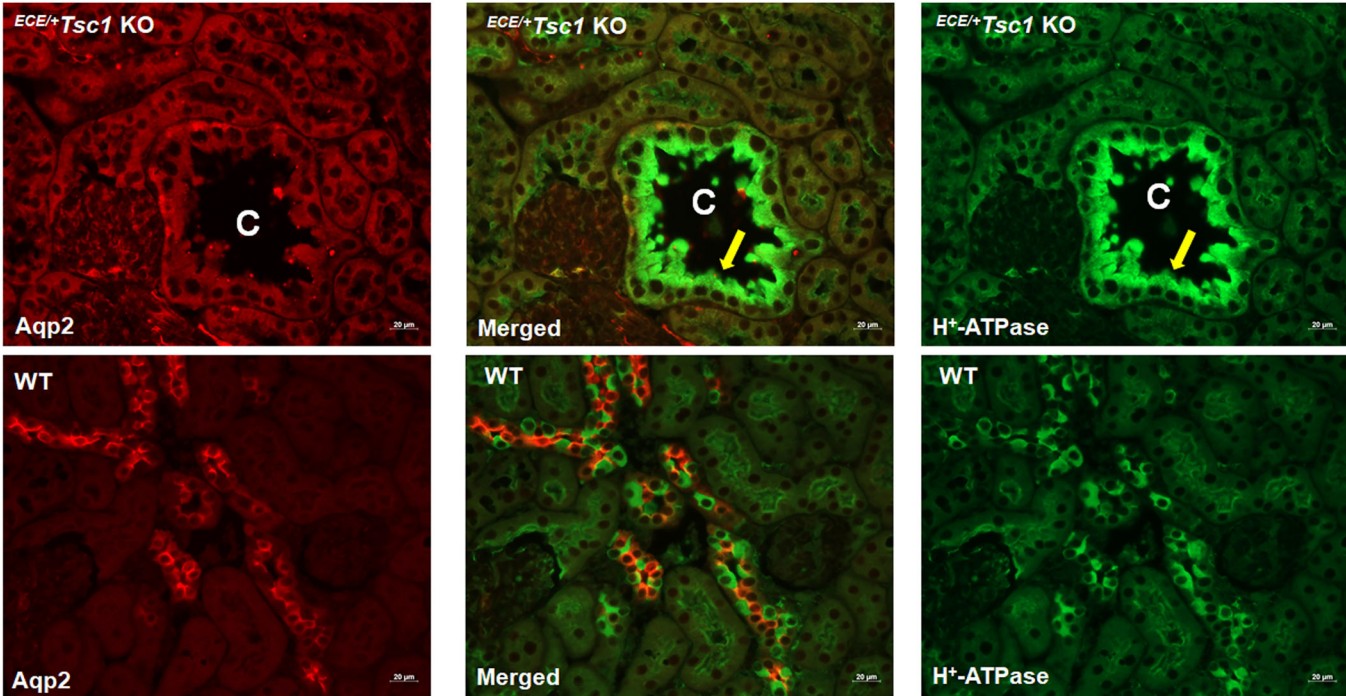

**Figure EV2. Localization of AQP2 and H⁺-ATPase in $^{ECE+/-}$Tsc1KO.**

Double immunofluorescence images of *Tsc1-KO* (upper row) and WT (lower row) mice stained with anti-AQP2 (red; left upper and lower panels) and H⁺-ATPase (green; right upper and lower panels) antibodies. A merged image illustrating co-localization of both antibodies is presented in the middle panel. White arrows point to basolateral AQP2 expression, while yellow arrows indicate apical H⁺-ATPase. "C" represents cysts. Scale bar equals 20 μm. Source data are available online for this figure.

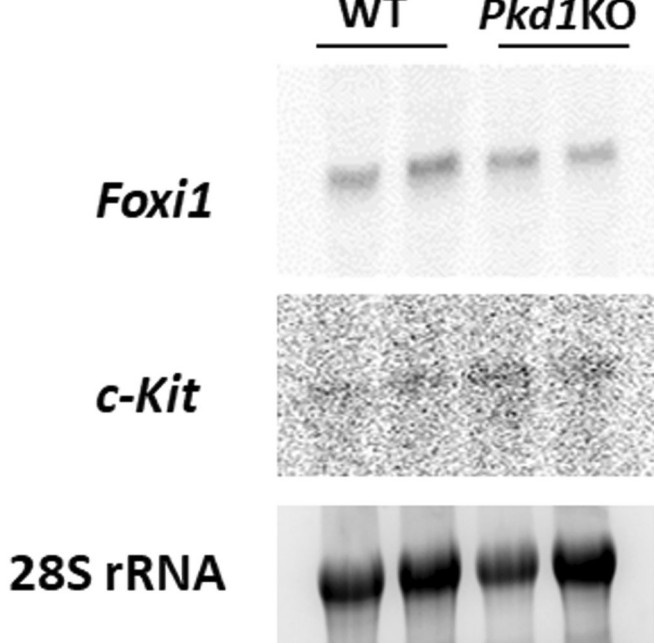

**Figure EV3. Comparison of expression of *Foxi1* and *c-Kit* in *Pkd1-KO* mice.**

Northern blot comparing *Foxi1* and *c-Kit* expression in *Pkd1-KO* vs. WT mice. Unlike *Tsc1-KO* mice, there is little to no expression of either of the two genes. Source data are available online for this figure.

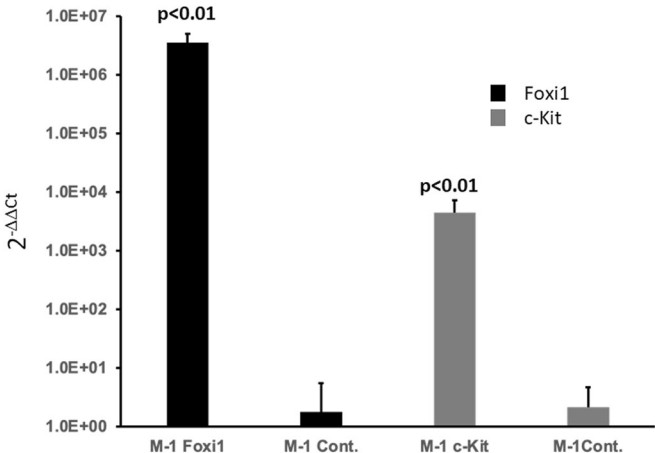

**Figure EV4.  Examination of *Foxi1* and *c-Kit* expression in transfected M-1 cells.**

M-1 collecting duct cell lines were stably transfected with *Foxi1, c-Kit* and control (empty) expression vectors. The expression of *Foxi1*(black) and *c-Kit* (gray) were compared in clonally purified stable transfectants. The studies represent values obtained from 5 clones for each transfected cell line ($n = 2$ replicas/clonal isolates). The "*p*" values were determined using two-tailed student t-test. Source data are available online for this figure.

   

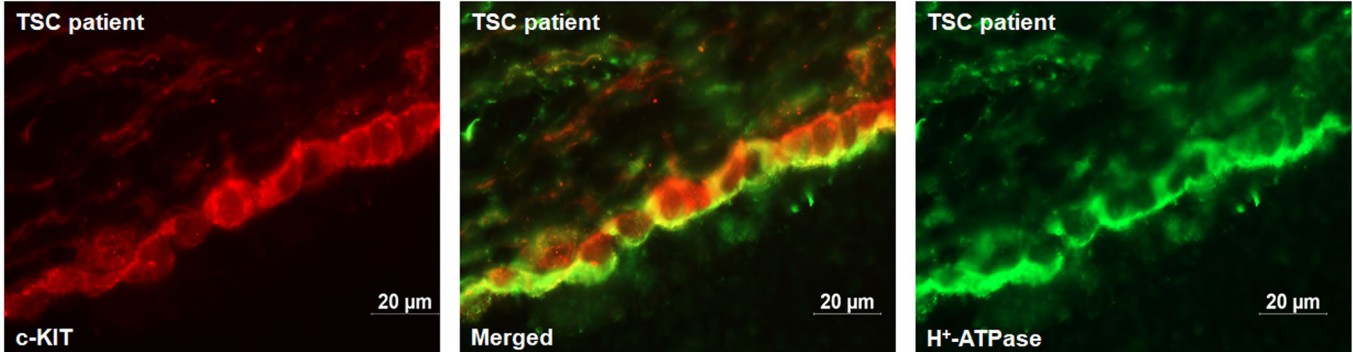

**Figure EV5. c-KIT and H⁺-ATPase expression in individuals with TSC.**

Similar to our TSC mouse models, individuals with TSC also have a basolateral localization of c-KIT (red; left panel) and apical distribution of H⁺-ATPase (green; right panel) in their cystic epithelium. Scale bar equals 20 μm. Source data are available online for this figure.

    

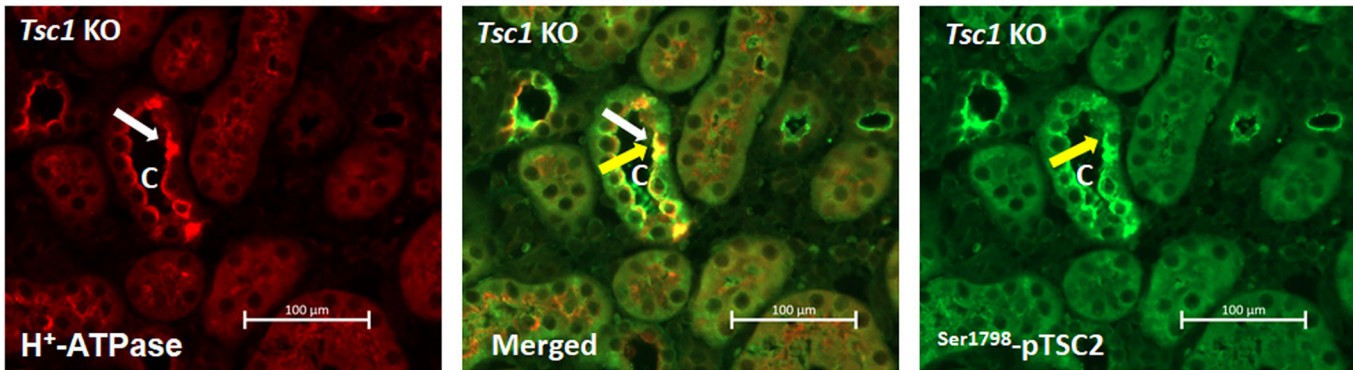

**Figure EV6. Localization of H⁺-ATPase and Ser1798-pTsc2 in *Tsc1-KO* mice.**

Double immunofluorescence images of apical H⁺-ATPase (red; left panel) and apical Ser1798-pTsc2 (green; right panel) in *Tsc1-KO* mice. A merged image illustrating co-localization of H⁺-ATPase and Ser1798-pTSC2 is presented in the middle panel. White arrows point to apical H⁺-ATPase expression, while yellow arrows indicate apical and subapical Ser1798-pTSC2 localization. "C" represents cysts. Scale bar equals 100 μm. Source data are available online for this figure.

 