## [Peer Review File · EMBO Molecular Medicine]

The Critical Role of the Proto-oncogene c-KIT in TSC Renal Cystogenesis

Kamyar Zahedi, Sharon Barone, Marybeth Brooks, Wenzheng Zhang, Jane Yu, Nathan Zaidman, and Manoocher Soleimani

Corresponding author(s): Manoocher Soleimani (MSoleimani@salud.unm.edu)

Review Timeline:

Submission Date:	1st Jul 25
Editorial Decision:	22nd Jul 25
Revision Received:	13th Oct 25
Editorial Decision:	27th Oct 25
Revision Received:	30th Oct 25
Editorial Decision:	11th Nov 25
Revision Received:	25th Nov 25
Accepted:	27th Nov 25

Editor: Zeljko Durdevic

Transaction Report:

22nd Jul 2025

Dear Dr. Soleimani,

Thank you for the submission of your manuscript to EMBO Molecular Medicine. We have now received feedback from the three reviewers who agreed to evaluate your manuscript. As you will see from the reports, all three referees are overall supportive of the study but also raise important concerns that should be addressed in a major revision. If you would like to discuss further the points raised by the referees, I am available to do so via email or video. Let me know if you are interested in this option.

We would welcome the submission of a revised version within three months for further consideration. Please let us know if you require longer to complete the revision.

I look forward to receiving your revised manuscript.

Yours sincerely,

Zeljko Durdevic

Zeljko Durdevic
Senior Editor
EMBO Molecular Medicine

We require:

- 1) A .docx formatted version of the manuscript text (including legends for main figures, EV figures and tables). Please make sure that the changes are highlighted to be clearly visible.
- 2) Individual production quality figure files as .eps, .tif, .jpg (one file per figure). For guidance, download the 'Figure Guide PDF': (<https://www.embopress.org/page/journal/17574684/authorguide#figureformat>).
- 3) A .docx formatted letter INCLUDING the reviewers' reports and your detailed point-by-point responses to their comments. As part of the EMBO Press transparent editorial process, the point-by-point response is part of the Review Process File (RPF), which will be published alongside your paper.
- 4) A complete author checklist, which you can download from our author guidelines (<https://www.embopress.org/page/journal/17574684/authorguide#submissionofrevisions>). Please insert information in the checklist that is also reflected in the manuscript. The completed author checklist will also be part of the RPF.
- 5) Please note that all corresponding authors are required to supply an ORCID ID for their name upon submission of a revised manuscript.
- 6) It is mandatory to include a 'Data Availability' section after the Materials and Methods. Before submitting your revision, primary datasets produced in this study need to be deposited in an appropriate public database, and the accession numbers and

database listed under 'Data Availability'. Please remember to provide a reviewer password if the datasets are not yet public (see <https://www.embopress.org/page/journal/17574684/authorguide#dataavailability>).

12) Author contributions: You will be asked to provide CRediT (Contributor Role Taxonomy) terms in the submission system. These replace a narrative author contribution section in the manuscript.

13) A Conflict of Interest statement should be provided in the main text.

14) Every published paper now includes a 'Synopsis' to further enhance discoverability. Synopses are displayed on the journal webpage and are freely accessible to all readers. They include a short stand first (maximum of 300 characters, including space) as well as 2-5 one-sentences bullet points that summarizes the paper. Please write the bullet points to summarize the key NEW findings. They should be designed to be complementary to the abstract - i.e. not repeat the same text. We encourage inclusion of key acronyms and quantitative information (maximum of 30 words / bullet point). Please use the passive voice. Please attach these in a separate file or send them by email, we will incorporate them accordingly.

15) Include a Reagents and Tools Table as part of the Methods section, which can be downloaded from our author guidelines (<https://www.embopress.org/page/journal/17574684/authorguide#structuredmethods>)

***** Reviewer's comments *****

Referee #1 (Comments on Novelty/Model System for Author):

In the present study, the authors proposed the proto-oncogene c-KIT as a novel player in the pathogenesis of TSC renal cystogenesis. To gain further insight into this complex topic, they generated new transgenic mice. The data are clear and have a significant clinical impact.

Referee #1 (Remarks for Author):

In the present study, the authors proposed that the proto-oncogene c-KIT plays a role in the pathogenesis of TSC renal cystogenesis. The data are clearly presented and explained. However, minor revisions would improve the quality of the paper, its scientific and clinical message.

1) FIG1E: the c-KIT phosphorylation level should be normalised to the total c-KIT abundance and not to actin, which is usually used to demonstrate equal protein loading. Therefore, a blot of c-KIT and a blot of phosphorylated c-KIT should be shown. The authors correctly showed this info in Fig. 7A. (total erk respect to phosphoerk etc...)

Histograms showing the densitometric analysis of the bands and the corresponding statistical studies should be included for each blot.

2) In addition to the Northern blot, could you also show blots in Figures 2, 3 and 5? This is an important point because mRNA levels do not always correspond to protein levels, possibly due to microRNA (miRNA) or circular RNA (circRNA), which may be important in abnormal proliferative cells.

3) The method used to measure cell proliferation in Fig. 4B is missing from the Materials and Methods section, and it is unclear how this was done. Is it cell proliferation or cell viability that reflects differences in cell senescence and death? Could you improve this by performing RT-PCR or blots with selective markers of cell proliferation? Fig. 4D shows green PCNA staining. However, the figure is unclear.

Referee #2 (Comments on Novelty/Model System for Author):

This is an excellent study. While it requires some revision and editing, I am confident the authors can address these points effectively. That said, I would caution against requesting extensive revisions that could take the authors six to eight months to complete.

Referee #2 (Remarks for Author):

In this manuscript by Zahedi et al., the authors report that the proto-oncogene receptor tyrosine kinase c-KIT is a critical driver of renal cystogenesis in Tuberous Sclerosis Complex (TSC). The authors used an orthogonal approach by combining genetic knockout models, transcriptomics, pharmacological inhibition and immunohistochemistry, to demonstrate that c-KIT is upregulated in cystic epithelia composed of A-intercalated cells (A-ICs) in various TSC mouse models and in human TSC kidney samples. Genetic ablation of c-Kit or pharmacologic inhibition by imatinib markedly reduced or eliminated cyst formation. Mechanistically, c-KIT activation enhances ERK1/2, AKT, and RSK1 signaling, which phosphorylates and inactivates TSC2, promoting mTORC1 activation and cyst proliferation. This elegant work is both mechanistic and translational, therefore worthy of publication in EMBO Mol Med. The strength of the paper lies in its comprehensive genetic and pharmacologic interrogation of c-KIT signaling in multiple models, and its potential therapeutic implication for treating TSC kidney disease. There are however

few issues that the authors need to address.

Comments

Integration with prior work

The authors must discuss their findings in the context of the study by Woodford et al. (EMBO J., 2017), which establishes TSC1 as a bona fide co-chaperone of Hsp90. That work demonstrates that TSC1 inhibits Hsp90 ATPase activity and facilitates the folding and stabilization of TSC2, thereby preventing its proteasomal degradation. Given this, the current finding in this manuscript that TSC2 is phosphorylated and inactivated, rather than degraded, during cystogenesis in TSC1-deficient kidneys raises a conceptual issue. The authors must address whether the observed TSC2 phosphorylation is occurring in lieu of TSC2 destabilization or whether impaired TSC1-Hsp90 chaperone function contributes to a noncanonical regulation of TSC2 stability and activity. In fact, Figure 6C supports (Woodford et al. EMBO J., 2017) work and demonstrates the presence of TSC1 phenocopies TSC2 protein levels. This point is essential for fully understanding the pathophysiology of TSC cystogenesis and the role of c-KIT in modulating chaperone dynamics.

- Figure 1B, 2B, 3B- The authors should quantify the expression of c-KIT and FOXI1 mRNA from Northern blots and compare them to the RNA-seq fold-changes to validate the transcriptomic results.
- Figure 1E- Total c-Kit levels is missing.
- Figure 6A: The mechanistic link between c-KIT signaling and TSC2 phosphorylation is central. The authors must show whether TSC2 and Tsc1 protein levels change alongside phosphorylation status.
- Figure 7C: The data suggest signaling attenuation in the dKO. However, the authors should include quantification across biological replicates. Do the data really suggest and confirm reduction in mTORC1?
- Figure 8A-D: Imatinib efficacy is promising. However, the authors must address off-target effects and whether other RTKs are expressed in these cystic cells.

Minor Comments:

- End of Introduction needs a short paragraph on the findings of the current paper.
- In the abstract, clarify whether ERK1/2, AKT, and RSK1 activation is dependent on c-KIT or merely correlated.
- Methods should indicate how tamoxifen administration was timed and verified in ECE/+Tsc1-KO mice.
- Figure legends lack statistical details-e.g., p-values, number of animals, error bars.
- Off target effect of Imatinib has not been discussed.

Referee #3 (Remarks for Author):

This work by Zahedi and colleagues examined the pathogenesis of cyst formation in the kidneys of TSC1 and TSC2 KO mice. RNAseq studies comparing kidneys of Tsc1-KO vs. WT and Tsc1/Foxi1-double-knockout identified c-Kit, as a transcript whose expression is significantly increased in Tsc1-KO mice. Overexpression of Foxi1 in kidney M-1 cells significantly increased c-Kit expression levels. Kidney cystogenesis was abolished in Tsc1-KO mice when c-Kit gene was also deleted (Tsc1/c-Kit-double-knockout mice). Treatment of Tsc1-KO mice with imatinib, a specific inhibitor of c-KIT, significantly diminished kidney cystogenesis. Renal cystogenesis was associated with ERK1/2, AKT, and RSK1-mediated phospho-inactivation of TSC2. In contrast, activation of ERK1/2, AKT, and RSK1, as well as phosphorylation of TSC2, was notably reduced in the kidneys of Tsc1/c-Kit-dKO mice. The authors propose that c-KIT is a crucial mediator of TSC renal cystogenesis and that inhibition of c-KIT may constitute a novel approach for the treatment of kidney cysts in TSC.

This excellent work constitutes extension of work previously done by this group. It is done in a rigorous manner and provides persuasive evidence for the involvement of c-KIT in the pathogenesis of cyst formation in the setting of TSC1 and/or TSC2 mutations, that is distinct from the pathway of cystogenesis in PKD. The work is nicely done and warrants publication.

There are a few minor points to address.

Please explain why it takes longer for TSC2 KO mice to develop cysts relative to TSC1 KO?

Please provide quantitation of TSC2 phosphorylations shown in Fig 7C.

X-Axis labels for Fig. 8D are not clearly visible. Please correct.

Page 4, line 124 should state "We compared the expression levels of c-Kit and Foxi1 mRNA in the kidneys of WT, Tsc1-KO, Tsc1/Foxi1-dKO, and Foxi1 KO mice."

Please provide more details on the M-1 cells.

REFEREE 1

The following is a point-by-point response to the concerns raised by the Referee.

Referee #1 (Comments on Novelty/Model System for Author):

In the present study, the authors proposed the proto-oncogene c-KIT as a novel player in the pathogenesis of TSC renal cystogenesis. To gain further insight into this complex topic, they generated new transgenic mice. The data are clear and have a significant clinical impact.

Response: We appreciate the informative and constructive comments from the Referee. We thank them for their kind comments regarding the novelty of our study.

Referee #1 (Remarks for Author):

In the present study, the authors proposed that the proto-oncogene c-KIT plays a role in the pathogenesis of TSC renal cystogenesis. The data are clearly presented and explained. However, minor revisions would improve the quality of the paper, its scientific and clinical message.

Response: We thank the Referee for their comments and suggestions regarding our manuscript.

Q1) FIG1E: the c-KIT phosphorylation level should be normalised to the total c-KIT abundance and not to actin, which is usually used to demonstrate equal protein loading. Therefore, a blot of c-KIT and a blot of phosphorylated c-KIT should be shown. The authors correctly showed this

Department of Internal Medicine
Division of Nephrology

info in Fig. 7A. (total erk respect to phosphoerk etc...). Histograms showing the densitometric analysis of the bands and the corresponding statistical studies should be included for each blot.

Response: A new western blot showing c-KIT and phosphorylated c-KIT(p-cKIT) with the associated histogram of the Image J densitometric analysis of the ratio of actin normalized to p-c-KIT and c-KIT is added to replace the previous Figure 1E.

Q2) In addition to the Northern blot, could you also show blots in Figures 2, 3 and 5? This is an important point because mRNA levels do not always correspond to protein levels, possibly due to microRNA (miRNA) or circular RNA (circRNA), which may be important in abnormal proliferative cells.

Response: Per our original discussion with the editor in July, Figs. 2, 3, and 5 of the original manuscript demonstrate the mRNA expression of c-Kit in three mouse models: 1) *Tsc1*-KO in principal cells using the inducible Cre mouse model, 2) global *Tsc2*^{+/-} mice, and 3) *Tsc1/c-Kit* double KO (dKO) mice. In addition, we have included immunofluorescence microscopy images using specific antibodies against c-KIT for all three mouse models. Extensive and strong c-KIT labeling on the basolateral membrane of cells lining the cysts reflects enhanced c-KIT protein abundance in congruence with mRNA expression levels (Figures 1B, 1D, 2B, 2C, 3B, 3C).

Some of these animals, including the *Tsc1* KO using inducible Cre mice (^{ECE/+}*Tsc1*-KO) or the *Tsc2*^{+/-} mice, require over 8 to 15 months to develop a significant kidney cyst burden (please see the resubmitted manuscript pages 4-5, lines 169-198). While we have RNA samples for gene expression studies or paraffin-embedded kidney sections for immunofluorescence labeling, harvesting new kidney tissues for protein preparation for western blotting will delay the resubmission of this manuscript by over a year. In addition, the results presented in Figure 1B vs Figure 1E indicate that there is a direct correlation between the expression of *c-Kit* mRNA and c-KIT protein levels.

We shared this information with Dr. Durdevic, the Editor of EMBO, and we hope the referee is satisfied with our data and the accompanying explanation.

Q3) The method used to measure cell proliferation in Fig. 4B is missing from the Materials and Methods section, and it is unclear how this was done. Is it cell proliferation or cell viability that reflects differences in cell senescence and death? Could you improve this by performing RT-PCR or blots with selective markers of cell proliferation? Fig. 4D shows green PCNA staining. However,

Department of Internal Medicine
Division of Nephrology

the figure is unclear.

Response: Regarding Figure 4B, we have now added the methodology used to measure cell proliferation in the Materials and Methods (lines 518-527). Crystal violet staining quantifies the density of cells in adherent cultures, where the increased cell density over time reflects increased cell density/cell proliferation (1, 2). In addition, the relationship between cell viability and cell proliferation is now discussed in the Results section (lines 226-227).

Concerning Figure 4D, we have now included new immunofluorescence images that clearly demonstrate the co-localization of c-KIT and PCNA to the same epithelial cells lining the cyst.

REFEREE 2

Referee #2 (Comments on Novelty/Model System for Author):

This is an excellent study. While it requires some revision and editing, I am confident the authors can address these points effectively. That said, I would caution against requesting extensive revisions that could take the authors six to eight months to complete.

Response: We thank the Referee for their kind comments regarding the novelty of our study.

Referee #2 (Remarks for Author):

In this manuscript by Zahedi et al., the authors report that the proto-oncogene receptor tyrosine kinase c-KIT is a critical driver of renal cystogenesis in Tuberous Sclerosis Complex (TSC). The authors used an orthogonal approach by combining genetic knockout models, transcriptomics, pharmacological inhibition and immunohistochemistry, to demonstrate that c-KIT is upregulated in cystic epithelia composed of A-intercalated cells (A-ICs) in various TSC mouse models and in human TSC kidney samples. Genetic ablation of c-Kit or pharmacologic inhibition by imatinib markedly reduced or eliminated cyst formation. Mechanistically, c-KIT activation enhances ERK1/2, AKT, and RSK1 signaling, which phosphorylates and inactivates TSC2, promoting mTORC1 activation and cyst proliferation. This elegant work is both mechanistic and translational, therefore worthy of publication in EMBO Mol Med. The strength of the paper lies in its comprehensive genetic and pharmacologic interrogation of c-KIT signaling in multiple models and its potential therapeutic implications for treating TSC kidney disease. There are, however, a few issues that the authors need to address.

Response: We thank the Referee for their encouraging comments and constructive suggestions regarding our manuscript.

Department of Internal Medicine
Division of Nephrology

Comments

Integration with prior work

Q1) The authors must discuss their findings in the context of the study by Woodford et al. (EMBO J., 2017), which establishes TSC1 as a bona fide co-chaperone of Hsp90. That work demonstrates that TSC1 inhibits Hsp90 ATPase activity and facilitates the folding and stabilization of TSC2, thereby preventing its proteasomal degradation. Given this, the current finding in this manuscript that TSC2 is phosphorylated and inactivated, rather than degraded, during cystogenesis in TSC1-deficient kidneys raises a conceptual issue. The authors must address whether the observed TSC2 phosphorylation is occurring in lieu of TSC2 destabilization or whether impaired TSC1-Hsp90 chaperone function contributes to a noncanonical regulation of TSC2 stability and activity. In fact, Figure 6C supports (Woodford et al. EMBO J., 2017) work and demonstrates the presence of TSC1 phenocopies TSC2 protein levels. This point is essential for fully understanding the pathophysiology of TSC cystogenesis and the role of c-KIT in modulating chaperone dynamics.

Response: We thank the referee for raising an intriguing possibility regarding the interaction of Tuberous Sclerosis Protein Complex (TSPC) with HSP90 and its impact on the stability of TSC2 in our model system of TSC renal cystic disease. We have now discussed the studies by Woodford et al (3) and Uhlmann et al (4) in the text in detail. These communications were shared with Dr. Durdevic, the editor, in July. As elaborated in the text, these *in vitro* studies utilize cultured HEK293 and MEF (*Tsc1*^{-/-}, *c-Abl*^{-/-}, etc.) cells to characterize the effects of HSP90 and HSP70 on TSC2 and the role of TSC1 as a co-chaperone of HSP90 (3). These studies also used a glial cell-specific *Tsc1*-cKO model (3, 4). The most significant points of these studies related to our work were that: 1) both TSC1 and TSC2 interact with HSP90; 2) inhibition HSP90 ATPase activity by TSC1 leads to a consequential enhancement of folding and stabilization of TSC2; and 3) TSC1, as a co-chaperone of HSP90, inhibits its ATPase function and promotes the folding and stability of its client proteins, including TSC2. The ablation of *Tsc1* in glial cells *in vivo* confirmed the role of TSC1 as a co-chaperone of HSP90 (Woodford et al. 2017) in TSC1 deficient glial cells(3, 4). These studies provide solid evidence for the crucial role of the TSC1-HSP90-TSC2 axis in regulating TSC activity.

There are several notable distinctions between the models used by (3) and our models of TSC renal cystogenesis that warrant discussion:

Department of Internal Medicine
Division of Nephrology

1. In our studies of kidney cystogenesis, we utilized an Aqp2 promoter-driven Cre transgene to knockout the *Tsc1* gene in kidney principal cells; however, the cells that make up the epithelium of kidney cysts are overwhelmingly A-intercalated cells that express both TSC1 and TSC2 (5, 6). In the model used by Woodford et al. (3, 4), the development of TSC-associated lesions is due to the expansion of glial cells that lack TSC1.
2. The changes in TSC1, TSC2, and Hsp90 proteins in HEK293 and MEF cells, as well as evidence of this axis in *Tsc1^{f/f}-Gfp Cre* mice (3), reflect a direct interaction between these molecules and the effect of disrupting this axis in glial cells, which drives their growth. However, the epithelial cells lining the kidney cysts in our TSC mouse model are distinct from the cells in which the *Tsc1* gene was ablated. In this model, the *Tsc1* gene is specifically ablated in principal cells; however, the lining of the cystic epithelium is almost entirely composed of A-intercalated cells (discussed in detail in our manuscript submitted to EMBO). The most important/interesting observation in this and other models of TSC cystogenesis, as well as in renal biopsies of individuals with TSC, is the expression of both TSC1 and TSC2 in the cystic epithelium and the absence of the loss of heterozygosity (LOH) that is the driver of TSC-associated hamartomas and astrocytes in *Tsc1^{f/f}-Gfp Cre* mice (discussed in detail in our manuscript submitted to EMBO). In brief, the renal cyst epithelia in TSC mouse models are composed of cells that express both TSC1 and TSC2. Therefore, the TSC1-HSP90-TSC2 axis is not affected, but TSCPC is inactivated, likely due to the phospho-inactivation of TSC2.
3. In every TSC mouse model (either the global *Tsc2^{+/-}*, *Tsc1* KO/renin 1 cre KO, *Tsc1/Aqp2* cre KO, or *Tsc2/Aqp2* cre KO) examined over the last decade, the kidney cyst epithelia are composed of A-intercalated cells that express both TSC1 and TSC2 proteins (all references pertaining to these models are cited and discussed in our current manuscript submitted to EMBO).
4. The presence of intact TSC1 and TSC2 in the cystic epithelium of TSC mouse models suggests that the TSC1-HSP90-TSC2 interaction remains intact and that TSC2 is properly folded, consistent with a stable conformation that is likely protected against degradation.

Department of Internal Medicine
Division of Nephrology

5. In conclusion, whereas the observation by Woodford et al. applies to TSC lesions caused by the LOH (e.g., hamartomas), signifying complete TSC1 deficiency, it does not apply to the cells lining the TSC kidney cysts and brain tubers, where LOH is not the driver of lesion formation. While our studies do not experimentally address the intactness of the TSC1-HSP90-TSC2 axis, the presence of TSC1 and TSC2 in the cystic epithelium suggests that the TSC1-HSP90-TSC2 interaction remains in place and that TSC2 is structurally intact.
6. The intact TSC2 in our model is targeted by kinases activated downstream of the c-KIT signal or other signals that stimulate these pathways, leading to its phosphorylation and subsequent deactivation.

We hope the referee is satisfied with this explanation and the delicate distinction between our studies in kidney cysts vs. the studies by Woodford et al (3).

- Figure 1B, 2B, 3B- The authors should quantify the expression of c-KIT and FOXI1 mRNA from Northern blots and compare them to the RNA-seq fold-changes to validate the transcriptomic results.

RESPONSE: The mRNA from Northern blots in Figures 1B, 2B, and 3B have now been quantified (via ImageJ), and related histograms have been included. It is not possible to do a direct comparison between the induction of mRNA from the Northern blots and the fold-changes from the RNA-Seq. The RNA in RNA-Seq is processed (enriched, fragmented, and reverse transcribed) and amplified; whereas, the RNA used in Northern blots is intact and does not undergo processing and amplification. While the results of RNA-seq and northern blot analysis strongly correlate, they do not exactly duplicate each other.

- Figure 1E- Total c-Kit levels is missing.

RESPONSE: A new image (Figure 1E) containing p-cKIT, total c-KIT and β -actin western blots and the corresponding Image J histogram has been added to replace the previous Figure 1E.

- Figure 6A: The mechanistic link between c-KIT signaling and TSC2 phosphorylation is central. The authors must show whether TSC2 and TSC1 protein levels change alongside phosphorylation status.

RESPONSE:

New images for TSC1 and TSC2 have been added to Fig. 6C which indicate that there are no

Department of Internal Medicine
Division of Nephrology

changes in the expression levels of either TSC1 or TSC2 between WT and *Tsc1* KO mice. However, there are significant changes in the phosphorylation states of TSC2 as was discussed in the original manuscript.

- Figure 7C: The data suggest signaling attenuation in the dKO. However, the authors should include quantification across biological replicates. Do the data really suggest and confirm reduction in mTORC1?

RESPONSE: We have now included new quantitative data regarding the activation of signaling pathways, phosphorylation of TSC2 (Figure 7C), and activation of mTORC1 (p-S6/S6) in WT, KO, and dKO mice (Figure 7D). Figure 7D (upper panel and lower panel) has now been added as a new figure and clearly indicates a significant reduction in mTORC1 activity in *Tsc1/c-Kit* dKO mice.

- Figure 8A-D: Imatinib efficacy is promising. However, the authors must address off-target effects and whether other RTKs are expressed in these cystic cells.

RESPONSE:

The potential role of the Imatinib Mesylate on non-c-Kit related RTKs, PDGFR, have been discussed elsewhere (Unachukwu et. al., 2023), and elaborated in the Discussion section (lines 374-385). In addition, the potential off target effects of Imatinib through its action on other RTKs and non-receptor TKs are delineated (lines 380-385).

Minor Comments:

- End of Introduction needs a short paragraph on the findings of the current paper.

RESPONSE:

A paragraph outlining the finding of this study has been added to the end of the Introduction section (lines 93-98).

- In the abstract, clarify whether ERK1/2, AKT, and RSK1 activation is dependent on c-KIT or merely correlated.

RESPONSE: We have a space constraint to add additional words to the abstract, but as can be seen in the manuscript we have clearly described the vital role of c-KIT in the signaling pathway leading to TSC cystogenesis.

Department of Internal Medicine
Division of Nephrology

- Methods should indicate how tamoxifen administration was timed and verified in ECE/+Tsc1-KO mice.

RESPONSE: Information concerning the dosage, route, and time of administration of tamoxifen has now been added to the Materials and Methods section (lines 441-445).

- Figure legends lack statistical details-e.g., p-values, number of animals, error bars.

RESPONSE: The Figure legends have now been updated to include statistical details, p-values, number of animals, and error bars.

- Off target effect of Imatinib has not been discussed.

RESPONSE:

As indicated, this concern has already been addressed in response to the Referee's earlier comment (please see our response to Referee's comments on Figure 8A-D; lines 380-384).

REFEREE 3

Referee #3 (Remarks for Author):

This work by Zahedi and colleagues examined the pathogenesis of cyst formation in the kidneys of TSC1 and TSC2 KO mice. RNAseq studies comparing kidneys of Tsc1-KO vs. WT and Tsc1/Foxi1-double-knockout identified c-Kit, as a transcript whose expression is significantly increased in Tsc1-KO mice. Overexpression of Foxi1 in kidney M-1 cells significantly increased c-Kit expression levels. Kidney cystogenesis was abolished in Tsc1-KO mice when c-Kit gene was also deleted (Tsc1/c-Kit-double-knockout mice). Treatment of Tsc1-KO mice with imatinib, a specific inhibitor of c-KIT, significantly diminished kidney cystogenesis. Renal cystogenesis was associated with ERK1/2, AKT, and RSK1-mediated phospho-inactivation of TSC2. In contrast, activation of ERK1/2, AKT, and RSK1, as well as phosphorylation of TSC2, was notably reduced in the kidneys of Tsc1/c-Kit-dKO mice. The authors propose that c-KIT is a crucial mediator of TSC renal cystogenesis and that inhibition of c-KIT may constitute a novel approach for the treatment of kidney cysts in TSC.

This excellent work constitutes extension of work previously done by this group. It is done in a rigorous manner and provides persuasive evidence for the involvement of c-KIT in the pathogenesis of cyst formation in the setting of TSC1 and/or TSC2 mutations, that is distinct from the pathway of cystogenesis in PKD. The work is nicely done and warrants publication.

RESPONSE: We greatly appreciate the kind assessment of our work by the referee.

Referee #3: There are a few minor points to address.

Department of Internal Medicine
Division of Nephrology

Please explain why it takes longer for TSC2 KO mice to develop cysts relative to TSC1 KO?

RESPONSE: *Tsc2^{+/-} mice are a different model of kidney cystogenesis driven by the loss of heterozygosity (LOH) in Tsc2 and generates a global phenotype that is distinct from our Tsc1 KO mice which caused by a specific deletion of the Tsc1 gene in principal cells. Despite the difference in the onset of the cystogenesis in these two models, both display a predominance of A-ICs and a paucity of principal cells in epithelial cells lining the cysts.*

Please provide quantitation of TSC2 phosphorylations shown in Fig 7C.

RESPONSE: *Quantifications of the Western blot in Figure 7C (calculated via ImageJ) has now been added, along with a histogram reflecting this analysis.*

X-Axis labels for Fig. 8D are not clearly visible. Please correct.

RESPONSE:

The X-Axis labels for Figure 8D (currently Fig 8E in the revised version) have now been enlarged for easy visibility.

Page 4, line 124 should state "We compared the expression levels of c-Kit and Foxi1 mRNA in the kidneys of WT, Tsc1-KO, Tsc1/Foxi1-dKO, and Foxi1 KO mice."

RESPONSE:

The suggested modification has been made (lines 140-142).

Please provide more details on the M-1 cells.

RESPONSE:

More information regarding the M-1 cells is provided in the Results (line 220) and Methods (lines 495-497) sections.

Best regards,

Manoocher Soleimani

Manoocher Soleimani, MD

Professor, Department of Medicine
University of New Mexico School of Medicine

Senior Clinician Scientist Investigator (2022-2030)
Veterans Administration Department

Department of Internal Medicine
Division of Nephrology

References

1. K. Chiba, K. Kawakami, K. Tohyama, Simultaneous evaluation of cell viability by neutral red, MTT and crystal violet staining assays of the same cells. *Toxicol In Vitro* **12**, 251-258 (1998).
2. M. Feoktistova, P. Geserick, M. Leverkus, Crystal Violet Assay for Determining Viability of Cultured Cells. *Cold Spring Harb Protoc* **2016**, pdb prot087379 (2016).
3. M. R. Woodford *et al.*, Tumor suppressor Tsc1 is a new Hsp90 co-chaperone that facilitates folding of kinase and non-kinase clients. *EMBO J* **36**, 3650-3665 (2017).
4. E. J. Uhlmann *et al.*, Astrocyte-specific TSC1 conditional knockout mice exhibit abnormal neuronal organization and seizures. *Ann Neurol* **52**, 285-296 (2002).
5. S. Barone *et al.*, Kidney intercalated cells and the transcription factor FOXi1 drive cystogenesis in tuberous sclerosis complex. *Proc Natl Acad Sci U S A* **118** (2021).
6. J. J. Bissler *et al.*, Tuberous sclerosis complex exhibits a new renal cystogenic mechanism. *Physiol Rep* **7**, e13983 (2019).

27th Oct 2025

Dear Dr. Soleimani,

Thank you for the submission of your revised manuscript to EMBO Molecular Medicine. I am pleased to inform you that we will be able to accept your manuscript pending the following final amendments:

1) Figures:

- During a standard image analysis, we detected potential duplications/reuse of images in different figures. Please review and clarify the following:

o In Fig 6A and 7A pERK and ERK are the same and correspond to 7A source data files.

o In Fig 1E-F, 5E and 6C Actin is the same, also in all source data files.

o In Fig 5E and 7D S6 and pS6 are the same also in the source data files but Actin is different.

o In Fig 7C and D Actin is the same.

o Please provide source data files of replicates for all Western and Northern blots.

- Please indicate molecular weights in all Western and Northern blots.

- Please submit all Appendix figures as EV Figures. EV figures should be uploaded as individual high-resolution files, with the legends in the manuscript text under the heading "Expanded View Figure Legends". Please update their callouts in the manuscript text. Please check "Author Guidelines" for more information.

<https://www.embopress.org/page/journal/17574684/authorguide#expandedview>

2) Author checklist: Please submit a complete checklist. <https://www.embopress.org/pb-assets/embo-site/EMBO%20Press%20Author%20Checklist-1642513524327.xlsx>

3) In the main manuscript file, please do the following:

- Please address all comments suggested by our data editors listed below:

o Figure legends:

1. Please note that the exact p values are not provided in the legends of figures 1E, F; 4A, B; 5E, 6A, C; 7A, C, D.

2. Please indicate the statistical test used for data analysis in the legends of figures 1E, 5E; 7A, C, D.

3. Please note that information related to n is missing in the legends of figures 7C, D.

4. Please note that the error bars are not defined in the legends of figures 1B, E, F; 2B, 3B, 4A, B; 5C, E; 6A, C; 7A, C, D.

- Add callouts for Fig 6B.

- Remove the information on the classification and the ORCID IDs.

- Remove all figures and only leave their legends at the end of the file.

- In Methods, provide the antibody dilutions that were used for each antibody.

- In Methods, provide sequences of northern blot probes.

- In Methods, a paragraph about TSC patient samples shown in Appendix Figure S5 is missing. Please provide the statement that informed consent was obtained from all human subjects and that the experiments conformed to the principles set out in the WMA Declaration of Helsinki and the Department of Health and Human Services Belmont Report.

- Please include structured Methods section that includes a Reagents and Tools Table (should be uploaded as a separate file) followed by a Methods and Protocols section. More information on how to adhere to this format as well as downloadable templates (.docx) for the Reagents and Tools Table can be found in our author guidelines:

<https://www.embopress.org/page/journal/17574684/authorguide#structuredmethods>

An example of a paper with Structured Methods can be found here:

<https://www.embopress.org/doi/full/10.1038/s44320-024-00037-6#sec-4>

- Indicate in legends exact n and exact p values, not a range, along with the statistical test used. To keep the figures "clear" some authors found providing an Appendix table Sx with all exact p-values preferable. You are welcome to do this if you want to.

- Author contributions: Please remove it from the manuscript and specify author contributions in our submission system. CRediT has replaced the traditional author contributions section because it offers a systematic machine-readable author contributions format that allows for more effective research assessment. You are encouraged to use the free text boxes beneath each contributing author's name to add specific details on the author's contribution. More information is available in our guide to authors:

<https://www.embopress.org/page/journal/17574684/authorguide#authorshipguidelines>

- In data availability statement please remove the current text. Data availability statement should contain information about raw data from large-scale datasets like RNA-seq that should be deposited in one of the relevant databases and made freely available prior the publication of the manuscript. please use the following format to report the accession number of your data:

[data type]: [full name of the resource] [accession number/identifier] ([doi or URL or identifiers.org/DATABASE:ACCESSION])

Please check "Author Guidelines" for more information.

<https://www.embopress.org/page/journal/17574684/authorguide#availabilityofpublishedmaterial>

- Correct the reference citation in the text and reference list. In the text, a reference should be cited by author and year of publication. Include a space between a word and the opening parenthesis of the reference that follows. In the reference list,

citations should be listed in alphabetical order. Where there are more than 10 authors on a paper, 10 will be listed, followed by "et al.". Please check "Author Guidelines" for more information.

<https://www.embopress.org/page/journal/17574684/authorguide#referencesformat>

4) Datasets: Rename the datasets to "Dataset EV1 - EV7 and add a legend with a short description to each file, in a separate tab/worksheet. Please update their callouts in the main text.

5) Synopsis: Every published paper now includes a 'Synopsis' to further enhance discoverability. Synopses are displayed on the journal webpage and are freely accessible to all readers. They include separate synopsis image and synopsis text.

- Synopsis image: Please provide a visual abstract as a high-resolution jpeg file 550 px-wide x 300-600 pixels high to illustrate your article.

- Synopsis text: Please provide a short standfirst (maximum of 300 characters, including space) as well as 2-5 one sentence bullet points that summarise the paper as a .doc file. Please write the bullet points to summarise the key NEW findings. They should be designed to be complementary to the abstract - i.e. not repeat the same text. We encourage inclusion of key acronyms and quantitative information (maximum of 30 words / bullet point). Please use the passive voice.

6) As part of the EMBO Publications transparent editorial process (see our Editorial at

<http://embomolmed.embopress.org/content/2/9/329>), EMBO Molecular Medicine will publish online a Review Process File (RPF) to accompany accepted manuscripts. This file will be published in conjunction with your paper and will include the anonymous referee reports, your point-by-point response and all pertinent correspondence relating to the manuscript. Let us know whether you agree with the publication of the RPF and as here, if you want to remove or not any figures from it prior to publication. Please note that the Authors checklist will be published at the end of the RPF.

7) Please provide a point-by-point letter INCLUDING my comments as well as the reviewer's reports and your detailed responses (as Word file).

I look forward to reading a new revised version of your manuscript as soon as possible.

Yours sincerely,

Zeljko Durdevic

Zeljko Durdevic
Senior Editor
EMBO Molecular Medicine

*** Instructions to submit your revised manuscript ***

1) a .docx formatted version of the manuscript text (including Figure legends and tables)

2) Separate figure files*

3) supplemental information as Expanded View and/or Appendix. Please carefully check the authors guidelines for formatting Expanded view and Appendix figures and tables at

<https://www.embopress.org/page/journal/17574684/authorguide#expandedview>

4) a letter INCLUDING the reviewer's reports and your detailed responses to their comments (as Word file).

5) The paper explained: EMBO Molecular Medicine articles are accompanied by a summary of the articles to emphasize the major findings in the paper and their medical implications for the non-specialist reader. Please provide a draft summary of your article highlighting

This may be edited to ensure that readers understand the significance and context of the research.

Please refer to any of our published articles for an example.

6) Author contributions: the contribution of every author must be detailed in a separate section.

7) EMBO Molecular Medicine now requires a complete author checklist

(<https://www.embopress.org/page/journal/17574684/authorguide>) to be submitted with all revised manuscripts. Please use the checklist as guideline for the sort of information we need WITHIN the manuscript. The checklist should only be filled with page numbers where the information can be found. This is particularly important for animal reporting, antibody dilutions (missing) and exact values and n that should be indicated instead of a range.

8) Every published paper now includes a 'Synopsis' to further enhance discoverability. Synopses are displayed on the journal webpage and are freely accessible to all readers. They include a short stand first (maximum of 300 characters, including space) as well as 2-5 one sentence bullet points that summarise the paper. Please write the bullet points to summarise the key NEW findings. They should be designed to be complementary to the abstract - i.e. not repeat the same text. We encourage inclusion of key acronyms and quantitative information (maximum of 30 words / bullet point). Please use the passive voice. Please attach these in a separate file or send them by email, we will incorporate them accordingly.

You are also welcome to suggest a striking image or visual abstract to illustrate your article. If you do please provide a jpeg file 550 px-wide x 300-600px high.

9) A Conflict of Interest statement should be provided in the main text

10) Please note that we now mandate that all corresponding authors list an ORCID digital identifier. This takes <90 seconds to complete. We encourage all authors to supply an ORCID identifier, which will be linked to their name for unambiguous name identification.

Currently, our records indicate that the ORCID for your account is 0000-0003-4909-4469.

Link Not Available

11) Include a Reagents and Tools Table as part of the Methods section, which can be downloaded from our author guidelines (<https://www.embopress.org/page/journal/17574684/authorguide#structuredmethods>)

Photos 400-800 DPI

*Additional important information regarding figures and illustrations can be found at

<https://bit.ly/EMBOPressFigurePreparationGuideline>. See also figure legend preparation guidelines:

<https://www.embopress.org/page/journal/17574684/authorguide#figureformat>

***** Reviewer's comments *****

Referee #2 (Remarks for Author):

The authors have addressed all my questions and concerns. This manuscript is ready for publication.

October 30, 2025

RE: EMM-2025-22130

Dear Dr. Durdevic:

Please see below our responses to the editorial suggestions in red font. We are replacing the existing files on the online submission platform with our corrected figures. We hope that all your enquiries are satisfactorily addressed.

1) Figures:

- During a standard image analysis, we detected potential duplications/reuse of images in different figures. Please review and clarify the following:
 - o In Fig 6A and 7A pERK and ERK are the same and correspond to 7A source data files. **Figures 6A and 7A were reviewed and the pERK and ERK images were corrected in Figure 6A.**
 - o In Fig 1E-F, 5E and 6C Actin is the same, also in all source data files. **New β -Actin images, using the same protein extracts, have now been added for Figs 5E and 6C, while the β -Actin image used for Figs 1E-F remains the same.**
 - o In Fig 5E and 7D S6 and pS6 are the same also in the source data files but Actin is different. **This image was a duplication in Figures 5 and 7. We have removed Figure 7D, but the image remains unchanged in Figure 5E. The Figure 7D description of S6 and pS6 have been removed from both the manuscript and figure legend.**
 - o In Fig7C and D Actin is the same. **Figure 7D has now been removed; however, the same β -Actin image had been used for Figures 5E, 7A, and 7C because they were the from the same extracts.**
 - o Please provide source data files of replicates for all Western and Northern blots. **All replicates for Western blots were provided as source data files. Northern blots were only performed once. Also, the original northern blot images have been provided as source data files.**
- Please indicate molecular weights in all Western and Northern blots. **Molecular weight markers are not generally used in Northern blots; however, the size estimation is based upon the location of the bands in comparison to 18s and 28s rRNA. We have now included the molecular weight markers for both the Western blots included as figures, as well as the Western blots in our source data.**
- Please submit all Appendix figures as EV Figures. EV figures should be uploaded as individual high-resolution files, with the legends in the manuscript text under the heading "Expanded View Figure Legends". Please update their callouts in the manuscript text. Please check "Author

Guidelines" for more information.

<https://www.embopress.org/page/journal/17574684/authorguide#expandedview>

The Appendix Figure Legends have now been added to the end of the manuscript and labeled as Figure EV1-6. They have also been renamed as Figure EV with the appropriate number in the manuscript and uploaded as individual high-resolution files.

2) Author checklist: Please submit a complete checklist. <https://www.embopress.org/pb-assets/embo-site/EMBO%20Press%20Author%20Checklist-1642513524327.xlsx>

This had already been included in our resubmission.

3) In the main manuscript file, please do the following:

- Please address all comments suggested by our data editors listed below:

o Figure legends:

1. Please note that the exact p values are not provided in the legends of figures 1E, F; 4A, B; 5E, 6A, C; 7A, C, D.

Exact p values have now been added to the legends of the above listed figures (except for 7D which has now been removed).

2. Please indicate the statistical test used for data analysis in the legends of figures 1E, 5E; 7A, C, D.

Statements have been added to the above figure legends indicating T-Test statistical analyses were conducted on Image J results.

3. Please note that information related to n is missing in the legends of figures 7C, D.

The information related to "n" has now been added to the legends for Figures 7C (Figure 7D has now been removed).

4. Please note that the error bars are not defined in the legends of figures 1B, E, F; 2B, 3B, 4A, B; 5C, E; 6A, C; 7A, C, D.

The following statement has now been added to the figure legends listed above: "Results are presented as average+ SD", except for Fig. 7D which has now been removed.

- Add callouts for Fig 6B.

The IHC staining is labeled and described in the figure legend and has now been referred to in the manuscript.

- Remove the information on the classification and the ORCID IDs.

This information has now been removed.

- Remove all figures and only leave their legends at the end of the file.

All figures have now been removed with only their legends remaining at the end of the file.

- In Methods, provide the antibody dilutions that were used for each antibody.

The antibody dilutions have now been added to the Methods section.

- In Methods, provide sequences of northern blot probes.

We have now provided the sequences of oligonucleotides utilized to generate our probes used for Northern blots.

- In Methods, a paragraph about TSC patient samples shown in Appendix Figure S5 is missing.

Please provide the statement that informed consent was obtained from all human subjects and that the experiments conformed to the principles set out in the WMA Declaration of Helsinki and the Department of Health and Human Services Belmont Report.

A statement has now been added to the Methods section describing that the patient sample was a gift from Dr. John Bissler and that it was obtained through an approved IRB with full

knowledge and consent of the patient.

- Please include structured Methods section that includes a Reagents and Tools Table (should be uploaded as a separate file) followed by a Methods and Protocols section. More information on how to adhere to this format as well as downloadable templates (.docx) for the Reagents and Tools Table can be found in our author guidelines:

<https://www.embopress.org/page/journal/17574684/authorguide#structuredmethods>

An example of a paper with Structured Methods can be found here:

<https://www.embopress.org/doi/full/10.1038/s44320-024-00037-6#sec-4>

We are now including a completed Reagents and Tools Table (based on the provided template) with our resubmission.

- Indicate in legends exact n and exact p values, not a range, along with the statistical test used. To keep the figures "clear" some authors found providing an Appendix table Sx with all exact p-values preferable. You are welcome to do this if you want to.

All figure legends have now listed exact "n" and p-values.

- Author contributions: Please remove it from the manuscript and specify author contributions in our submission system. CRediT has replaced the traditional author contributions section because it offers a systematic machine-readable author contributions format that allows for more effective research assessment. You are encouraged to use the free text boxes beneath each contributing author's name to add specific details on the author's contribution. More information is available in our guide to authors:

<https://www.embopress.org/page/journal/17574684/authorguide#authorshipguidelines>

Author contributions have now been removed from the manuscript.

- In data availability statement please remove the current text. Data availability statement should contain information about raw data from large-scale datasets like RNA-seq that should be deposited in one of the relevant databases and made freely available prior the publication of the manuscript. please use the following format to report the accession number of your data:

[data type]: [full name of the resource] [accession number/identifier] ([doi or URL or identifiers.org/DATABASE:ACCESSION])

Please check "Author Guidelines" for more information.

<https://www.embopress.org/page/journal/17574684/authorguide#availabilityofpublishedmaterial>

Due to the United States government shutdown, we are unable to upload our datasets at this time; however, we have included the datasets in our submission so readers can easily access them.

- Correct the reference citation in the text and reference list. In the text, a reference should be cited by author and year of publication. Include a space between a word and the opening parenthesis of the reference that follows. In the reference list, citations should be listed in alphabetical order. Where there are more than 10 authors on a paper, 10 will be listed, followed by "et al.". Please check "Author Guidelines" for more information.

<https://www.embopress.org/page/journal/17574684/authorguide#referencesformat>

The references have now been corrected in the manuscript based on the author guide and information listed above.

4) Datasets: Rename the datasets to "Dataset EV1 - EV7 and add a legend with a short description to each file, in a separate tab/worksheet. Please update their callouts in the main text.

Datasets have now been renamed throughout the manuscript based on the comment above.

5) Synopsis: Every published paper now includes a 'Synopsis' to further enhance discoverability. Synopses are displayed on the journal webpage and are freely accessible to all readers. They include separate synopsis image and synopsis text.

- Synopsis image: Please provide a visual abstract as a high-resolution jpeg file 550 px-wide x 300-600 pixels high to illustrate your article.

- Synopsis text: Please provide a short standfirst (maximum of 300 characters, including space) as well as 2-5 one sentence bullet points that summarise the paper as a .doc file. Please write the bullet points to summarise the key NEW findings. They should be designed to be complementary to the abstract - i.e. not repeat the same text. We encourage inclusion of key acronyms and quantitative information (maximum of 30 words / bullet point). Please use the passive voice.

We had included a Synopsis (Graphical Abstract) in our resubmission.

6) As part of the EMBO Publications transparent editorial process (see our Editorial at <http://embomolmed.embopress.org/content/2/9/329>), EMBO Molecular Medicine will publish online a Review Process File (RPF) to accompany accepted manuscripts. This file will be published in conjunction with your paper and will include the anonymous referee reports, your point-by-point response and all pertinent correspondence relating to the manuscript. Let us know whether you agree with the publication of the RPF and as here, if you want to remove or not any figures from it prior to publication. Please note that the Authors checklist will be published at the end of the RPF.

7) Please provide a point-by-point letter INCLUDING my comments as well as the reviewer's reports and your detailed responses (as Word file).

This has now been uploaded to the EMBO submission site. We could not find any additional reviewers' reports on the submission site. Our previous responses have already been uploaded in our second cover letter (dated 9/29/25), but we are also including them here for easy reference:

11th Nov 2025

Dear Dr. Soleimani,

Thank you for the submission of your revised manuscript to EMBO Molecular Medicine. I am about to accept your paper, however there are few issues that require your attention.

1) Figures: Thank you for clarifying and correcting Western blot images. However, several Western blots still share the same β -Actin loading control. This is acceptable only if the images originate from the same blot. You stated that the same β -Actin images are used in different figures because they were from the same protein extracts. Please note that this is not the same as being from the same blot, because even identical extract samples can be loaded differently on separate gels etc. Please clarify whether the Western blots sharing the same β -Actin loading control originate from the same membrane that was re-probed with different antibodies. If this is the case, please add the following sentence in the "Protein Extraction and Western Blot Analysis" paragraph: Western blot images that share the same β -Actin loading control - Figure 1E and 1F, Figure 6A and 6C as well as Figure 5E, 7A and 7C - originate from the same Western blot membranes.

2) Correct the callout Appendix Dataset S1-4 to Datasets EV1-4.

3) Please clarify why the RNA-Seq analysis description has been removed from the Methods, while the corresponding data remain in the Results section. If you want to present these data, a full description of the RNA-Seq methodology and analysis must be reinstated in the Methods section. If you decide to remove the data, we will need to editorially re-evaluate the suitability of the manuscript for EMBO Molecular Medicine.

4) Data availability statement is mandatory. We understand the current issues with depositing data at NCBI. However, as we are mandating the deposition of unprocessed data from large scale experiments like RNA-Seq we would like you to use alternative repositories such as EMBL-EBI or DDBJ to deposit your data that should be made freely available prior the publication.

5) Synopsis:

- Synopsis image: Please remove the text from the visual abstract and upload the image as a high-resolution jpeg file 550 px-wide x 300-600 pixels high. The image should convey the main message of the study using minimal text.

- Synopsis text: Please upload the synopsis text as a separate .doc file. Synopsis should contain a short standfirst (maximum of 300 characters, including space) as well as 2-5 one sentence bullet points that summarise the paper. Please write the bullet points to summarise the key NEW findings. They should be designed to be complementary to the abstract - i.e. not repeat the same text. We encourage inclusion of key acronyms and quantitative information (maximum of 30 words / bullet point). Please use the passive voice.

6) Please response to the following query: As part of the EMBO Publications transparent editorial process (see our Editorial at <http://embomolmed.embopress.org/content/2/9/329>), EMBO Molecular Medicine will publish online a Review Process File (RPF) to accompany accepted manuscripts. This file will be published in conjunction with your paper and will include the anonymous referee reports, your point-by-point response and all pertinent correspondence relating to the manuscript. Let us know whether you agree with the publication of the RPF and as here, if you want to remove or not any figures from it prior to publication. Please note that the Authors checklist will be published at the end of the RPF.

7) Please provide a point-by-point response to my comments as Word file.

I look forward to receiving the revised version of your manuscript as soon as possible.

Yours sincerely,

Zeljko Durdevic

Zeljko Durdevic
Senior Editor
EMBO Molecular Medicine

November 11, 2025

RE: EMM-2025-22130

Dear Dr. Durdevic:

Please see our responses to your editorial suggestions in red font below:

- 1) Figures: Thank you for clarifying and correcting Western blot images. However, several Western blots still share the same β -Actin loading control. This is acceptable only if the images originate from the same blot. You stated that the same β -Actin images are used in different figures because they were from the same protein extracts. Please note that this is not the same as being from the same blot, because even identical extract samples can be loaded differently on separate gels etc. Please clarify whether the Western blots sharing the same β -Actin loading control originate from the same membrane that was re-probed with different antibodies. If this is the case, please add the following sentence in the "Protein Extraction and Western Blot Analysis" paragraph: Western blot images that share the same β -Actin loading control - Figure 1E and 1F, Figure 6A and 6C as well as Figure 5E, 7A and 7C - originate from the same Western blot membranes.
The following sentence has now been added to lines 450-452: " Western blot images that share the same β -Actin loading control - Figure 1E and 1F, Figure 6A and 6C as well as Figure 5E, 7A and 7C - originate from the same Western blot membranes." The reason we are using the same B-Actin is because the membranes can only be used a maximum of 3 times.
- 2) Correct the callout Appendix Dataset S1-4 to Datasets EV1-4.
Line 197 has now been corrected to "Datasets EV1-4."
- 3) Please clarify why the RNA-Seq analysis description has been removed from the Methods, while the corresponding data remain in the Results section. If you want to present these data, a full description of the RNA-Seq methodology and analysis must be reinstated in the Methods section. If you decide to remove the data, we will need to editorially re-evaluate the suitability of the manuscript for EMBO Molecular Medicine.
The RNA-Seq analysis description has now been added in the Methods section (lines 447-457).
- 4) Data availability statement is mandatory. We understand the current issues with depositing data at NCBI. However, as we are mandating the deposition of unprocessed data from large scale experiments like RNA-Seq we would like you to use alternative repositories such as EMBL-EBI or DDBJ to deposit your data that should be made freely available prior the publication.
As per guidelines for funding from the United States Department of Defense, NIH, and the Department of Veterans Affairs, we are mandated to use the Gene Expression Omnibus (GEO) as a repository of our data. Please let us know if you need additional clarification on this issue.
- 5) Synopsis:
 - Synopsis image: Please remove the text from the visual abstract and upload the image as a high-resolution jpeg file 550 px-wide x 300-600 pixels high. The image should convey the main message of the study using minimal text.

- Synopsis text: Please upload the synopsis text as a separate .doc file. Synopsis should contain a short standfirst (maximum of 300 characters, including space) as well as 2-5 one sentence bullet points that summarise the paper. Please write the bullet points to summarise the key NEW findings. They should be designed to be complementary to the abstract - i.e. not repeat the same text. We encourage inclusion of key acronyms and quantitative information (maximum of 30 words / bullet point). Please use the passive voice.

The Synopsis image has now been saved as a high-quality jpeg based on the dimensions provided and has been separated from the Synopsis Text. These are uploaded as two separate files.

- 6) Please response to the following query: As part of the EMBO Publications transparent editorial process (see our Editorial at <http://embomolmed.embopress.org/content/2/9/329>), EMBO Molecular Medicine will publish online a Review Process File (RPF) to accompany accepted manuscripts. This file will be published in conjunction with your paper and will include the anonymous referee reports, your point-by-point response and all pertinent correspondence relating to the manuscript. Let us know whether you agree with the publication of the RPF and as here, if you want to remove or not any figures from it prior to publication. Please note that the Authors checklist will be published at the end of the RPF.

We agree to the publication of the RPF.

Warm regards,

Manoocher Soleimani, MD
Professor, Department of Medicine
University of New Mexico School of Medicine

Senior Clinician Scientist Investigator (2022-2030)
Department of Veterans Health

27th Nov 2025

Dear Dr. Soleimani,

We are pleased to inform you that your manuscript is accepted for publication and is now being sent to our publisher to be included in the next available issue of EMBO Molecular Medicine.

Zeljko Durdevic
Senior Editor
EMBO Molecular Medicine
